# Design of a recombinant asparaginyl ligase for site-specific modification using efficient recognition and nucleophile motifs
Jiabao Tang [1,2,3,4,5,6,8], Mengling Hao[1,2,3,4,5,6,8], Junxian Liu[1,2,3,4,5,6], Yaling Chen[1,2,3,4,5,6], Gulimire Wufuer[1,2,3,4,5,6], Jie Zhu[7], Xuejie Zhang[1,2,3,4,5,6], Tingquan Zheng[1,2,3,4,5,6], Mujin Fang[1,2,3,4,5,6], Shiyin Zhang[1,2,3,4,5,6], Tingdong Li [1,2,3,4,5,6] ✉, Shengxiang Ge [1,2,3,4,5,6] ✉, Jun Zhang[1,2,3,4,5,6] & Ningshao Xia[1,2,3,4,5,6]

Asparaginyl ligases have been extensively utilized as valuable tools for site-specific bioconjugation or surface-modification. However, the application is hindered by the laborious and poorly reproducible preparation processes, unstable activity and ambiguous substrate requirements. To address these limitations, this study employed a structure-based rational approach to obtain a high-yield and high-activity protein ligase called OaAEP1-C247A-aa55-351. It was observed that OaAEP1-C247A-aa55-351 exhibits appreciable catalytic activities across a wide pH range, and the addition of the $Fe^{3+}$ metal ion effectively enhances the catalytic power. Importantly, this study provides insight into the recognition and nucleophile peptide profiles of OaAEP1-C247A-aa55-351. The ligase demonstrates a higher recognition ability for the "Asn-Ala-Leu" motif and an N-terminus "Arg-Leu" as nucleophiles, which significantly increases the reaction yield. Consequently, the catalytic activity of OaAEP1-C247A-aa55-351 with highly efficient recognition and nucleophile motif, "Asn-Ala-Leu" and "Arg-Leu" under the buffer containing $Fe^{3+}$ is 70-fold and 2-fold higher than previously reported OaAEP1-C247A and the most efficient butelase-1, respectively. Thus, the designed OaAEP1-C247A-aa55-351, with its highly efficient recognition and alternative nucleophile options, holds promising potential for applications in protein engineering, chemo-enzymatic modification, and the development of drugs.

Protein modification and ligation play pivotal roles in biochemistry, protein engineering, and drug development[1–4]. Chemical modification of proteins at the amine-, carboxyl-, and thiol-groups of proteins represents the primary approach[1,3,5–8] and is widely utilized. However, this chemical synthesis process is highly reliant on the specific amino acid residues, which exhibit considerable variation among different proteins[1,9,10]. Consequently, substantial case-by-case optimization becomes necessary. Additionally, the conjugates can mask epitopes and cause protein inactivation. In contrast, bioconjugation through enzyme-catalytic ligation offers numerous advantages, displaying immense potential in protein synthesis and modification engineering. These advantages include mild reaction conditions, highly specific site-targeted modification, extensive versatility, and the ability to preserve protein bioactivity with ease[1,10,11].

[1]State Key Laboratory of Vaccines for Infectious Diseases, School of Public Health, Xiamen University, 361102 Xiamen, China. [2]National Institute of Diagnostics and Vaccine Development in Infectious Diseases, School of Public Health, Xiamen University, 361102 Xiamen, China. [3]National Innovation Platform for Industry-Education Integration in Vaccine Research, School of Public Health, Xiamen University, 361102 Xiamen, China. [4]NMPA Key Laboratory for Research and Evaluation of Infectious Disease Diagnostic Technology, School of Public Health, Xiamen University, 361102 Xiamen, China. [5]Department of Laboratory Medicine, School of Public Health, Xiamen University, 361102 Xiamen, China. [6]Xiang An Biomedicine Laboratory, 361102 Xiamen, China. [7]Jiangsu Key Laboratory of Advanced Catalytic Materials and Technology, School of Petrochemical Engineering, Changzhou University, 213164 Changzhou, China. [8]These authors contributed equally: Jiabao Tang, Mengling Hao. ✉e-mail: litingdong@xmu.edu.cn; sxge@xmu.edu.cn

Recent advancements have been achieved in the development and application of enzymatic bioconjugation[12,13], including ligase[14–20], transferases[21], inteins[22–26], transpeptidases[27–31]. Notably, the peptide/protein ligases currently identified are primarily composed of Sortase A[16,18,27,32] and peptide asparaginyl ligases (PALs), such as butelase-1[14,33], OaAEP1b[34], HeAEP3[35], OaAEP3-5[36], VyPAL1/2[17,37], which facilitate the formation of peptide bonds[14,33,34,38–40]. Sortase A (srtA) harbors an N-terminal signal peptide, a membrane anchor motif, and a core catalytically active domain[41]. LPXTG motif is the recognition sequence of srtA and allows site-specific conjugation. However, its application is limited by the relatively long recognition sequence (LPXTG) and poor ligating efficiency due to reversible reactions. PALs are expressed as inactive zymogens, containing a vacuole-targeted signal peptide, N-terminal pro-domain, catalytic core domain, and C-terminal cap domain that covers the active site[42]. Auto-activation occurred at acidic conditions and led to the cleavage of both the pro- and cap domains at the N- and C-termini of the catalytic core and the release of mature active enzyme[34,35,37,40,43]. The active PALs catalyze transpeptidation at the Asn residue of a short Asn-Xaa1-Xaa2 tripeptide motif at around neutral pH. PALs address the challenge of long recognition sequences seen in Sortase A[33]. Among PALs, butelase-1, extracted from the tropical cyclotide-producing plant *Clitoria ternatea*, has proven to be the most efficient ligase with broad applications in the food and biopharmaceutical fields[14,33]. Butelase-1 specifically recognizes the Asx-His-Val tripeptide sequence at the C terminus of the protein. The enzyme breaks the peptide bond between Asx and His, generating a residue that links to the amino-terminal residue of another protein, resulting in the formation of an Asx-Xaa and completing the protein-linking process[4,14,44,45]. Compared to previous peptide ligases, butelase-1 demonstrates superior efficiency in ligation and faster catalytic speed[12,17,33,46]. Additionally, the catalytic ability of butelase-1 far exceeds its hydrolysis capacity[33]. This solves the shortcomings of the slow catalytic rate and reversible reaction of Sortase A. As a result, butelase-1 opens up possibilities in biotechnology, protein engineering, and various other fields. Recent advances have enabled the recombinant expression of butelase-1; however, it is still challenged by unsatisfactory yields, undesirable catalytic efficiency, and complicated, time-consuming expression and purification processes[46–48].

OaAEP1 is an asparaginyl endopeptidase (AEP) evolutionarily related to butelase-1 derived from the plant *Oldenlandia affinis*[34]. Recently, the crystal structure of OaAEP1 has been resolved[40], providing valuable insights into its functional properties. OaAEP1 consists of 474 amino acids, including a signal peptide, a core domain, and a cap structure. The core domain and the cap domain are connected by a linker, with the cap covering the active site of the core domain, rendering OaAEP1 inactive. Full-length OaAEP1 can be expressed in *E. coli* and is self-activated under acidic conditions (pH 3.4–4.0) to release the cap[40]. The Cystine at position 247 may serve as an important channel for nucleophiles to enter. The OaAEP1-C247A mutant, with a cysteine-to-alanine substitution at position 247, displays superior catalytic efficiency compared to the wild type and is even comparable to butelase-1[40]. However, the application was still limited by the cumbersome purification procedure, unstable activation efficiency, poorly reproducible preparation processes at acidic conditions, and ambiguous substrate requirements. Further research efforts should focus on simplifying the recombinant production of AEP ligases and sharpening the understanding of AEP ligase substrate selectivity.

In this study, we aimed to develop an efficient chemo-enzymatic site-specific conjugation strategy for protein engineering, surface modification, and drug development. Using a structure-based rational approach, we succeeded in obtaining a robust and high-yielding ligation enzyme, OaAEP1-C247A-aa55-351, which overcame the challenges associated with reduced enzymatic activity caused by harsh acid activation and low productivity resulting from complicated operating procedures. Furthermore, we conducted a comprehensive investigation into the substrate selectivity spectrum and enzymatic activity features of OaAEP1-C247A-aa55-351, resulting in a substantial improvement in the yield of the ligation products.

The OaAEP1-C247A-aa55-351, with highly efficient substrate recognition and nucleophilic motif, exhibited 70 times higher catalytic activity compared to the previously reported OaAEP1-C247A. This enhanced activity broadens the potential applications of OaAEP1-C247A-aa55-351 in biochemical synthesis, protein engineering, chemo-enzymatic modification, and drug development.

## Results and discussion
### Design of a protein ligase based on the structure of OaAEP1-C247A

According to the structure basis, the 271 amino acid long core domain of OaAEP1 is essential to exert its ligation activity (Fig. 1a). But the catalytic cleft is entirely shielded by the cap domain (Fig. 1a). So, the release of cap domain is greatly warranted for protein or peptide substrates to get access to the active site (Fig. 1b, c). Currently, it remains unclear whether aa24-54 and aa326-351 play crucial roles in the ligase function. Therefore, four truncations were designed: OaAEP1-C247A-aa24-351, OaAEP1-C247A-aa55-351, OaAEP1-C247A-aa24-325, OaAEP1-C247A-aa55-325 (Fig. 1d, nucleotide sequences of the four truncations see Supplementary Methods in the Supplementary Information). These truncations were expressed in *E. coli* as fusion proteins with N-terminal TrxA-Tag and C-terminal His-tag and were purified using a cobalt-based IMAC column (Supplementary Fig. 1a, b). The proteins with high levels of purity were obtained and confirmed by SDS–PAGE analysis (Fig. 1e and Supplementary Fig. 1c). Taking the molecular weight of the fusion protein TrxA, his-tag, linker, and that of the target protein into account, the size of each recombinant OaAEP1-C247A truncations agrees with the expected molecular weight: OaAEP1-C247A-aa24–351, 55 kDa; OaAEP1-C247A-aa55-351, 51 kDa; OaAEP1-C247A-aa24-325, 52 kDa; OaAEP1-C247A-aa55-325, 48 kDa.

### Ligation activity of active OaAEP1

In order to evaluate the ligation activity of the four truncated proteins, we synthesized the model substrate Pep133-NGL. This substrate was created by adding the recognition motif of OaAEP1 ligase "Asn-Gly-Leu" to the C-terminal of peptide aa133-147 of human cytomegalovirus pp65 protein. Another peptide having N-terminal residues, "Gly-Leu" (GL- (biotin-labeled) peptide), was utilized as the nucleophile. The enzymatic conversion product was analyzed via enzyme-linked immunosorbent assay (ELISA) using an antibody against peptide aa133-147 (Fig. 2a). All four truncated proteins exhibited enzymatic ligating activity, with OaAEP1-C247A-aa55-351 performing the best (Fig. 2b, Supplementary Data 2). Therefore, OaAEP1-C247A-aa55-351 was further investigated.

To elucidate whether only peptide containing the reported recognition site "Asn-Gly-Leu" can be recognized and ligated by OaAEP1-C247A-aa55-351, other peptides of pp65, Pep445, and Pep133 were used to replace Pep133-NGL as substrate, and the results showed that none of these peptides could be ligated with the biotinylated GL peptide (Fig. 2c, Supplementary Data 2), which illustrates the substrate specificity of OaAEP1-C247A-aa55-351.

To determine whether auto-cleavage occurred during the expression and purification of OaAEP1-C247A-aa55-351, amino acid sequencing was performed to determine its C-terminal boundary. Purified OaAEP1-C247A-aa55-351 proteins were separated by SDS–PAGE, followed by in-gel digestion with trypsin, and subjected to amino acid sequencing by LC-MS/MS. The theoretical amino acid sequence of the recombinant OaAEP1-C247A-aa55-351 (Supplementary Fig. 1a) was used as a template to obtain the sequence. Auto cleavage was not observed according to the result of LC–MS/MS. The amino acid sequence was consistent with the theoretical sequence (Fig. 2d).

### Optimization of the ligating conditions of OaAEP1-C247A-aa55-351

The ligating efficiency may be greatly affected by the buffer conditions, specifically the pH and the presence of metal ions. Previous studies have

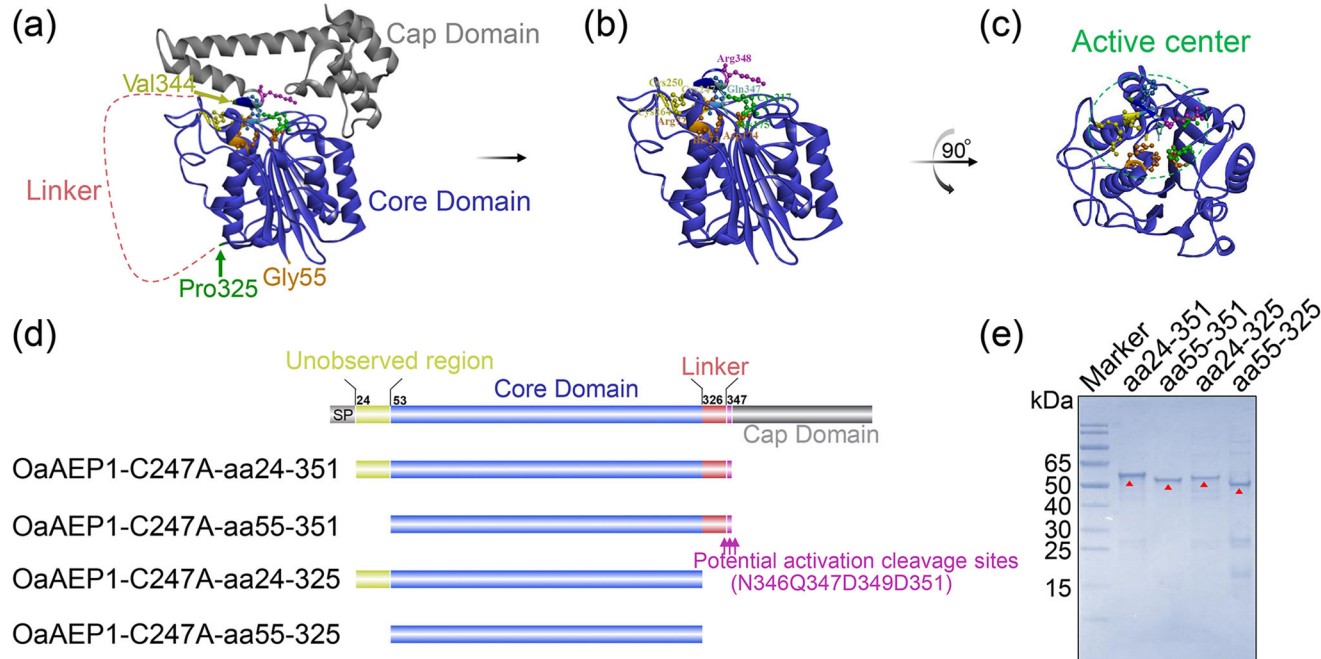

**Fig. 1 | Design and purification of protein ligase. a** Crystal structure of OaAEP1 as zymogenic form (from PDB code: 5H0I). **b** Active form of OaAEP1 by removal of C-terminal cap domain. **c** The top view of activated OaAEP1 is obtained by 90° rotation around the vertical as presented in (**b**). S1 pocket (Active center) with active site residues displayed as scaled balls, sticks, and labeled. **d** Schematic representation of the truncated activated OaAEP1. SP signal peptide; Unobserved region means the region unobserved in crystal structure. **e** SDS–PAGE analysis of fractions (1 μg) after purification of the OaAEP1-C247A truncations. Red triangles indicate the target bands.

reported that OaAEP1-C247A displays high catalytic activity towards P1-Asn substrates at a neutral pH level (pH 7.0)[40]. To determine the pH preference of OaAEP1-C247A-aa55-351, pH scanning assays were conducted in various pH buffers ranging from pH 4.0 to 9.6 (Fig. 3a). The ligating products were tested using ELISA as previously described in Fig. 2a. The results showed that OaAEP1-C247A-aa55-351 displayed ligation activity between pH 4.0 and 8.0, with the highest ligating efficiency observed between pH 7.0 and 7.2 (Fig. 3b, Supplementary Data 2). The enzyme exhibited reduced activity in acidic or basic buffer conditions, but certain activity was observed at pH 4.0 and 8.0 (Fig. 3b, Supplementary Data 2). These findings suggest that the pH preference of OaAEP1-C247A-aa55-351 is similar to the original OaAEP1-C247A.

Metal ion, such as $Mg^{2+}$, $Mn^{2+}$, $Ca^{2+}$, $Zn^{2+}$, $Fe^{3+}$, can facilitate various enzymatic reactions[49–51]. Considering that no available studies have examined the role of metal ions on the ligase activity of OaAEP1, we tested several common metal ions, including $Ca^{2+}$, $Mg^{2+}$, $K^+$, $Fe^{2+}$, $Fe^{3+}$, $Mn^{2+}$, $Zn^{2+}$ (Fig. 3c). The results demonstrated that the first three metal ions had no significant effect on the ligation activity of OaAEP1-C247A-aa55-351, whereas the other four ions facilitate the ligating activity of OaAEP1-C247A-aa55-351, among which $Fe^{3+}$ performs the best (Fig. 3d, Supplementary Data 2).

### Substrate and nucleophile specificity of OaAEP1-C247A-aa55-351

Previous studies have reported the C-terminal tripeptide recognition motif of OaAEP1, Asn-Gly-Leu (NGL, P1–P1'–P2'), is broken between P1 and P1' during the catalytic process. This occurs as the catalytic cysteine of OaAEP1 performs a nucleophilic attack and leads to the formation of the acyl-enzyme intermediate between the catalytic cysteine and the P1 Asn residue[34,37,40,43,52]. Then, the amine group of the N-terminal Gly-Leu (GL)-based nucleophile peptide (P1''–P2'') attack onto the formed unstable acyl-enzyme intermediate breaks the transient thioester bond and releases the ligating product from the catalytic cysteine[37,43,52]. Previous studies have demonstrated the recognized and

nucleophilic motif for peptide ligation by OaAEP1, but the results were not so consistent and adequate[53,54]. To further validate and augment the profiles of recognition and nucleophile peptides, five positions, namely P1–P1'–P2' and P1''–P2'', were substituted with Ala, Gln, Phe, Asp, Lys, respectively, representing amino acids categorized as non-polar aliphatic, polar neutral, aromatic, acidic and basic amino acids (Fig. 4a). In accordance with prior findings, P1 Asn exhibited relatively stringent requirements, as substitution of P1 with other amino acids resulted in the unidentifiable and unligated peptides (Fig. 4b, Supplementary Data 2). On the other hand, the positions of P1' or P2' were found to be less restricted. Interestingly, When P1' G was replaced by A or K, the ligating efficiency was further improved, with P1' A demonstrating the best performance overall (Fig. 4b, Supplementary Fig. 2b and Supplementary Data 2). Though P1' Q or F and P2' A or F remained recognizable, their ligation efficiencies were lower (Fig. 4b, Supplementary Data 2). In terms of the nucleophile peptide, P2'' displayed greater stringency compared to P1'', and it exhibited a nucleophilic attack capability only when P2'' L was substituted with F and not with the other four amino acids (Fig. 4c, Supplementary Data 2). While P1'' exhibited more flexibility, P1'' F or P1'' K outperformed the original P1'' G, with P'' K showing the highest efficiency (Fig. 4c, Supplementary Data 2). Furthermore, to determine the optimal "partner", P1' A and P1'' K are substituted with amino acids possessing similar characteristics and subsequently paired (Fig. 4d), and the combination of Asn-Ala-Leu (NAL, P1–P1'–P2') with Arg-Leu (RL, P1''–P2'') achieved the highest ligation efficiency (Fig. 4e, Supplementary Fig. 2c and Supplementary Data 2). Moreover, the presence of position P1'' at the N-terminus of the peptides or proteins was essential for successful ligation (Fig. 4c, Supplementary Data 2). The superior ligating activity of Asn-Ala-Leu (NAL, P1–P1'–P2') paired with Arg-Leu (RL, P1''–P2'') could be attributed to two factors. Firstly, it may be attributed to the stronger electrophilic nature of Ala in comparison to Gly, making it more susceptible to nucleophilic attack. Secondly, Arg exhibits a stronger electropositive cavity than Gly, facilitating nucleophilic attack.

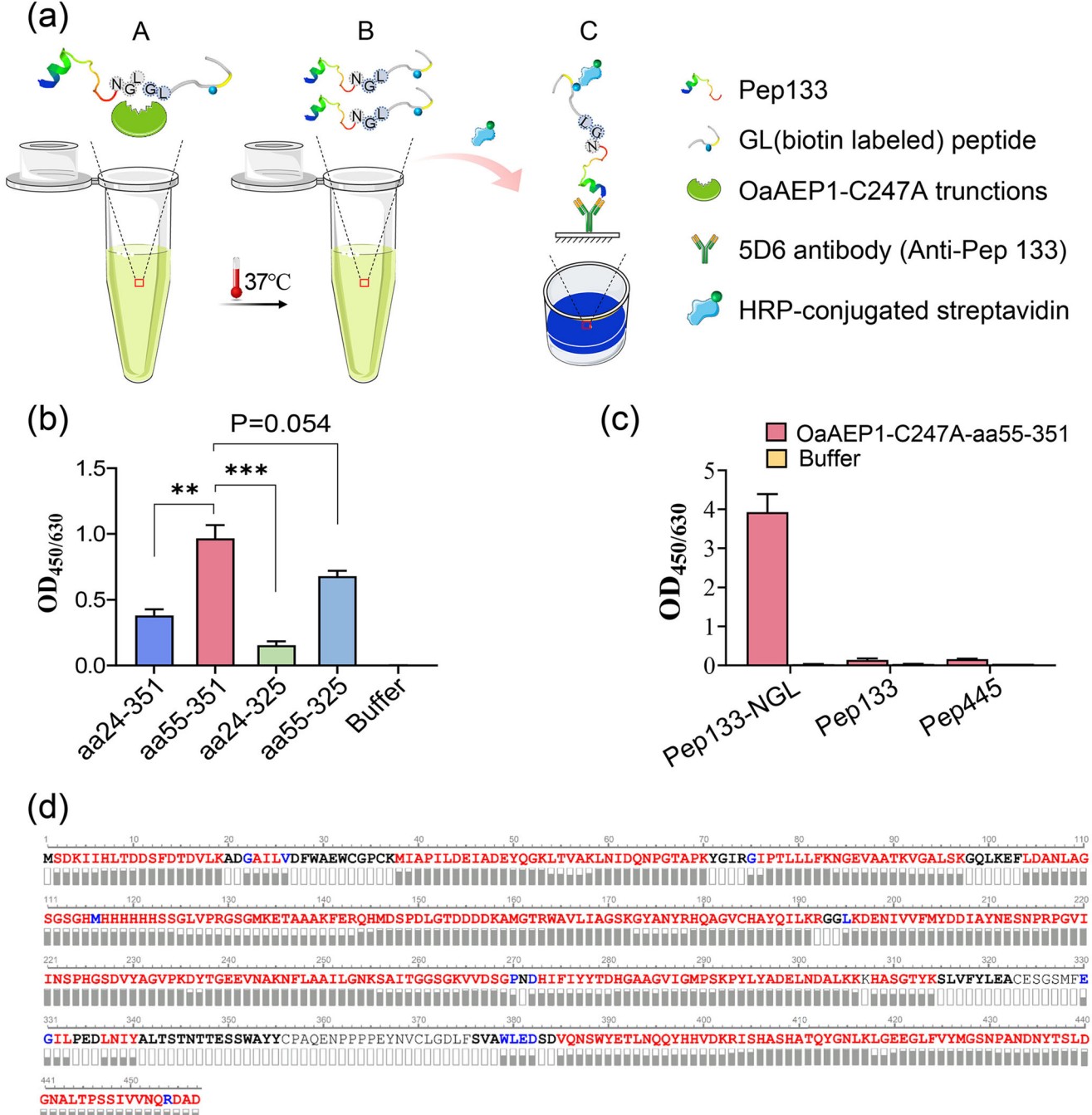

**Fig. 2 | Enzymatic ligating activity evaluation of the four truncated protein.**
**a** Schematic representation of ligating assay and the evaluation mode of the reaction products after ligating by the truncated OaAEP1. "NGL", "GL" based peptides and truncated OaAEP1 mixed at proper buffer (A) and ligating at 37 °C. Ligating products (B) were subjected to capture by antibodies against "NGL" based peptide and detected by HRP-conjugated streptavidin (C). **b** The ligating activity of the different truncated OaAEP1-C247A. Pep133-NGL (3 μM) and GL-(biotin-labeled) peptide (30 μM) were ligated by the different truncated OaAEP1-C247A (0.5 μM) at 37 °C in PBS for 30 min, and the ligating products were detected by ELISA showed in (**a**).

**c** The specificity of OaAEP1-C247A-aa55–351 was verified by peptides of CMV pp65 free of Asn-Gly-Leu. Pep133-NGL/Pep133/Pep445: 5 μM; GL-(biotin-labeled) peptide: 50 μM; OaAEP1-C247A-aa55–351: 1.5 μM. Multiple groups were compared using ordinary one-way ANOVA. \*\*$p < 0.01$, \*\*\*$p < 0.001$. **d** Sequence coverage view of OaAEP1-C247A-aa55-351 protein by LC–MS/MS analysis on an Orbitrap Eclipse Tribrid mass spectrometer. The amino acids are colored in red, blue and black, which represents three different levels of amino acid confidence from high to low. The assays are performed as $n = 3$ biologically independent experiments, and the bars represent the mean and standard deviation (SD).

Based on the optimal substrate of Asn-Ala-Leu (P1–P1'–P2') and nucleophile peptide of Arg-Leu (P1"–P2"), the effect of $Fe^{3+}$ was further validated, and the optimum concentration of $Fe^{3+}$ was investigated. The result demonstrated that a moderate concentration of $Fe^{3+}$ enhances the yield of the ligating product, with 1 mM proving to be the most effective (Fig. 4f, Supplementary Data 2). Given that the catalytic residue is cystine, we introduced oxi-reductive addictives to examine the oxidation-reduction effects of metal ions on enhancing the ligase activity. However, both the oxidizing agent ($H_2O_2$) and the reducing agent (β-mercaptoethanol) appeared to have a detrimental effect on the ligation activity of OaAEP1-C247A-aa55–351, and the former seemed worse (Supplementary Fig. 2d). Therefore, the role of metal ions

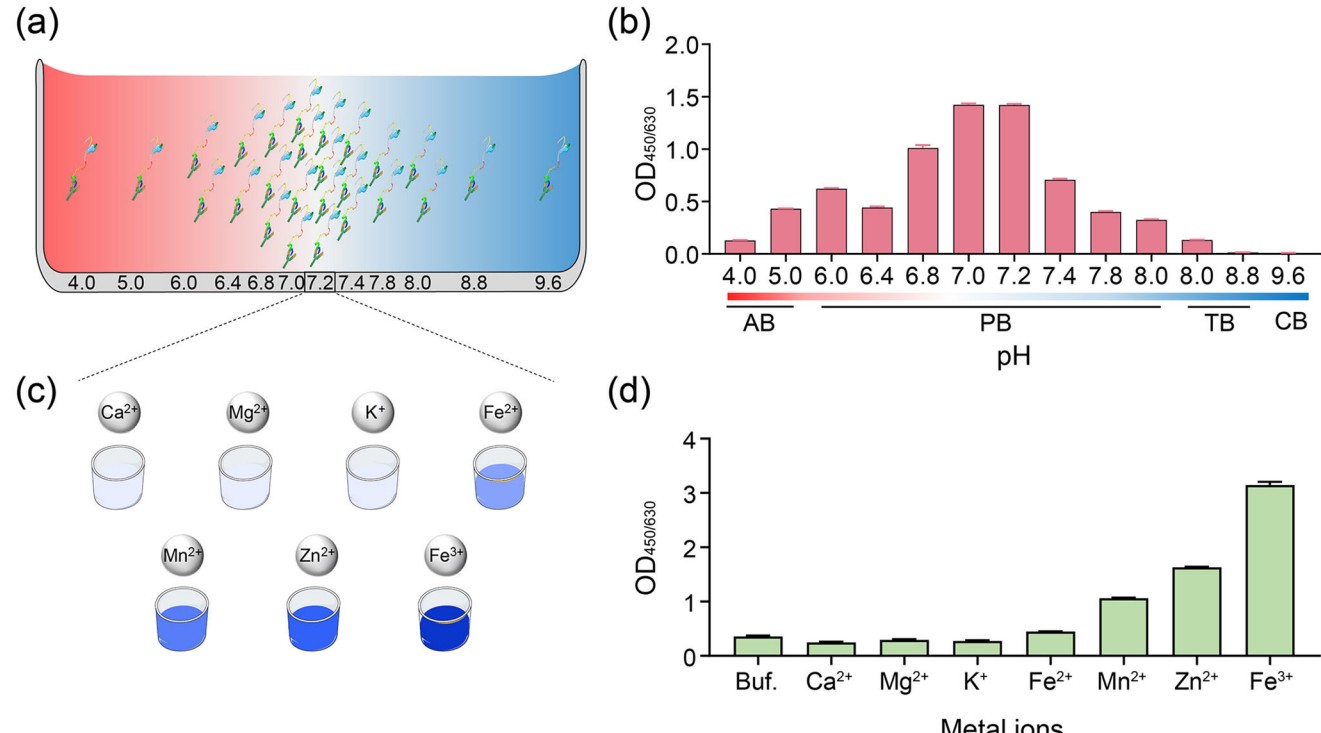

**Fig. 3 | pH and metal ions preference profile of OaAEP1-C247A-aa55–351.**
Schematic diagram of ligating substrates and nucleophiles under buffer different pH
(**a**) and metal ions (**c**). AB: 50 mM sodium acetate buffer; PB: 20 mM phosphate
buffer; TB: 50 mM Tris–HCl buffer. **b** The ligating activity of Pep133-NGL (2 μM)
and GL-(biotin-labeled) peptide (10 μM), which ligated in buffers with different pH
by the OaAEP1-C247A-aa55-351 ligase (0.5 μM) at 37 °C for 30 min. **d** The effect of

different metal ions (100 μM) in enhancing the ligating efficiency of OaAEP1-
C247A-aa55-351. Buf.: Buffer (phosphate buffer, pH 7.2); $Ca^{2+}$: $CaCl_2$; $Mg^{2+}$: $MgCl_2$;
$K^+$: KCl; $Fe^{2+}$: $FeSO_4$; $Mn^{2+}$: $MnCl_2$; $Zn^{2+}$: $ZnSO4$; $Fe^{3+}$: $FeCl_3$. The assays are per-
formed as $n = 3$ biologically independent experiments, and the bars represent the
mean and SD.

necessitates further exploration for a comprehensive understanding in
future studies.

## OaAEP1-C247A-aa55-351 mediated highly efficient peptide and protein labeling

To investigate the kinetics of OaAEP1-C247A-aa55-351 in the ligating
substrate and nucleophiles, a Förster resonance energy transfer (FRET)
ligation assay was conducted. This assay utilized the pep133-NXL
(X = Ala or Gly, FAM labeled) peptide as the substrate and the XL
peptide (X = Arg or Gly, Dabcyl labeled) as the nucleophile, enabling
real-time monitoring of ligated product formation (Fig. 5a, b). As the
substrate and the nucleophile were ligated, the close physical proximity
of the two fluorophores facilitated energy transfer from FAM to Dabcyl,
resulting in reduced FAM ($\lambda_{ex}$ = 510 nm) fluorescence. The decrease in
relative fluorescence unit (RFU) over time demonstrated successful
ligation of the substrates and nucleophiles (Fig. 5c, Supplementary
Data 2). The catalytic rate exhibited a rapid rise with increasing substrate
concentration, reaching a plateau thereafter (Fig. 5d). Notably, the $K_{cat}$/
$K_m$ value of OaAEP1-C247A-aa55-351 towards the substrate recogni-
tion motif "Asn-Ala-Leu" and nucleophiles "Arg-Leu" under the con-
dition of $Fe^{3+}$ was approximately 24 times higher than OaAEP1-C247A-
aa55-351 towards the previously reported "Asn-Gly-Leu" and "Gly-
Leu", and 70 times higher than the ultrafast variant AEP(Cys247Ala), as
well as 2 times higher than butelase-1[14], the most efficient ligase reported
to date (Fig. 5d, e). Interestingly, OaAEP1-C247A-aa55-351 exhibited
markedly enhanced catalytic activity compared to butelase-1 or AEP(-
Cys247Ala). This improvement can primarily be attributed to the mild
purification process, which circumvents the reduction in enzyme
activity associated with acidic activation, as well as the enhanced ligating
yield benefiting from more efficient recognition and nucleophile motifs,
as well as the incorporation of metal ions.

To further assess the product yield following ligation by OaAEP1-
C247A-aa55-351, we synthesized 50 amino acids long peptide C50-NAL
and RL- (biotin-labeled) polyXXK peptide to observe noticeable changes
between the ligation products and the substrates prior to ligation. After
ligation by OaAEP1-C247A-aa55-351 under the buffer with $Fe^{3+}$, the pro-
ducts were subjected to analysis by SDS–PAGE. The results indicated that
over 90% of C50-NAL was ligated with RL- (biotin-labeled) polyXXK
peptide (Supplementary Fig. 3a, b), demonstrating that OaAEP1-C247A-
aa55-351 is capable of producing high yields of products.

Protein labeling is a powerful tool in the field of biomedical and bio-
technology. To assess the ability of OaAEP1-C247A-aa55-351 to facilitate
well-folded protein labeling, we expressed and purified recombinant trun-
cated nucleocapsid protein (aa1-258) of severe acute respiratory syndrome
coronavirus 2 (SARS-CoV-2) with a C-terminal "Asn-Ala-Leu", designated
as rtNP-NAL (Supplementary Fig. 3c). Nucleophiles in the form of "Arg-
Leu" (RL- (biotin-labeled) peptide) were used. The resulting product was
detected using ELISA, similar to Fig. 2a, utilizing the antibody against rtNP,
17H11, coated on the microplate. The results demonstrated that the biotin-
labeled peptide could be conjugated to rtNP-NAL (Supplementary Fig. 3d),
indicating that properly folded recombinant proteins are equally capable of
site-specific modification with the assistance of OaAEP1-C247A-aa55-351.

## Conclusion

Given the potential of AEPs in biotechnology and peptide or protein
engineering applications, a concerted effort has been made to enhance
performance, efficiency, and accessibility. Herein, we explored a design
approach on the basis of the recently determined crystal structure of the
OaAEP1-C247A catalytic domain and constructed a ligase, OaAEP1-
C247A-aa55-351, which eliminates the complicated processes of acid acti-
vation and further purification. As a result, a ligase with specific protein
ligating activity was obtained through one-step expression and purification.

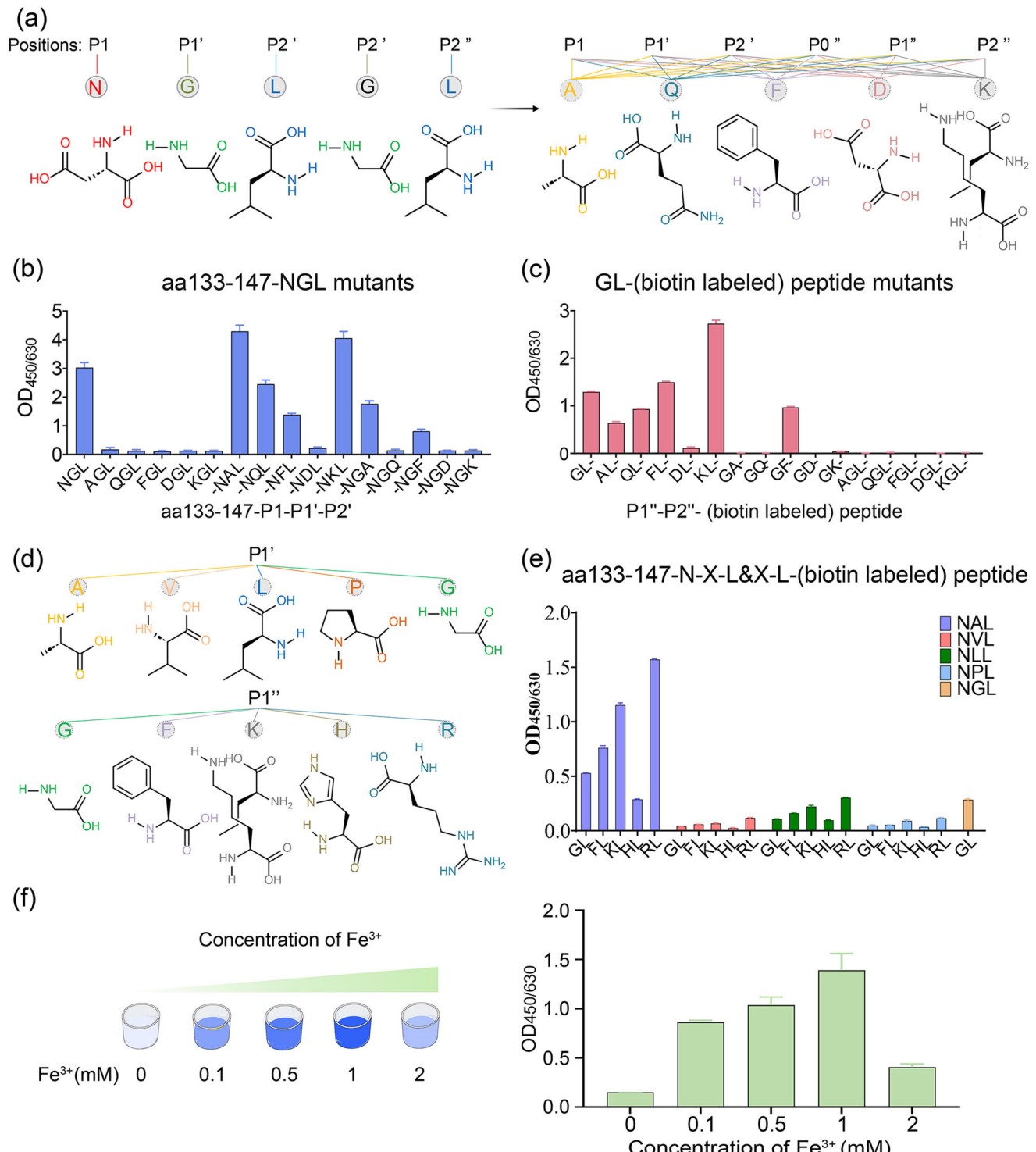

**Fig. 4 | Hydrolysis and nucleophile profile for peptide ligation catalyzed by OaAEP1-C247A-aa55-351. a** Schematic representation of amino acid structures of the five positions of P1–P1'–P2' and P1"–P2", and five kinds of representative amino acid including non-polar aliphatic, polar neutral, aromatic, acidic and basic amino acids. **b, c** The ligating efficiency of different Pep133-P1–P1'–P2' mutants to GL-(biotin-labeled) peptide (**b**) or Pep133-NAL to different P1"–P2"– (biotin-labeled) peptide mutants (**c**). P0" indicates the position in front of P1". **d** Schematic representation of amino acid structures of the same group of position P1' Ala and P1" Lys, i.e., non-polar aliphatic (Val, Leu, Pro) and basic amino acids (His, Arg). **e** The best partner of hydrolysis and nucleophile peptide of OaAEP1-C247A-aa55–351 (Pep133-NXL links with *L- (biotin-labeled) peptide (X = A or V or L or P or G, *G or F or K or H or R)). **f** The impact of concentrations of $Fe^{3+}$ on the ligation activity of OaAEP1-C247A-aa55-351. The assays are performed as $n = 3$ biologically independent experiments, and the bars represent the mean and SD.

Subsequently, it was demonstrated that the utilization of the more effective and efficient recognition motif "Asn-Ala-Leu" and alternative nucleophiles "Arg-Leu" under the catalytic ligation of OaAEP1-C247A-aa55-351 leads to higher yields. Moreover, $Fe^{3+}$ with suitable concentration was proved to enhance the ligation efficiency of OaAEP1-C247A-aa55-351. The $K_{cat}/K_m$ of OaAEP1-C247A-aa55-351 toward superior substrate recognition motif "Asn-Ala-Leu" and nucleophiles "Arg-Leu" is 24 times higher than the previously reported "Asn-Gly-Leu" and "Gly-Leu", and 70 times higher

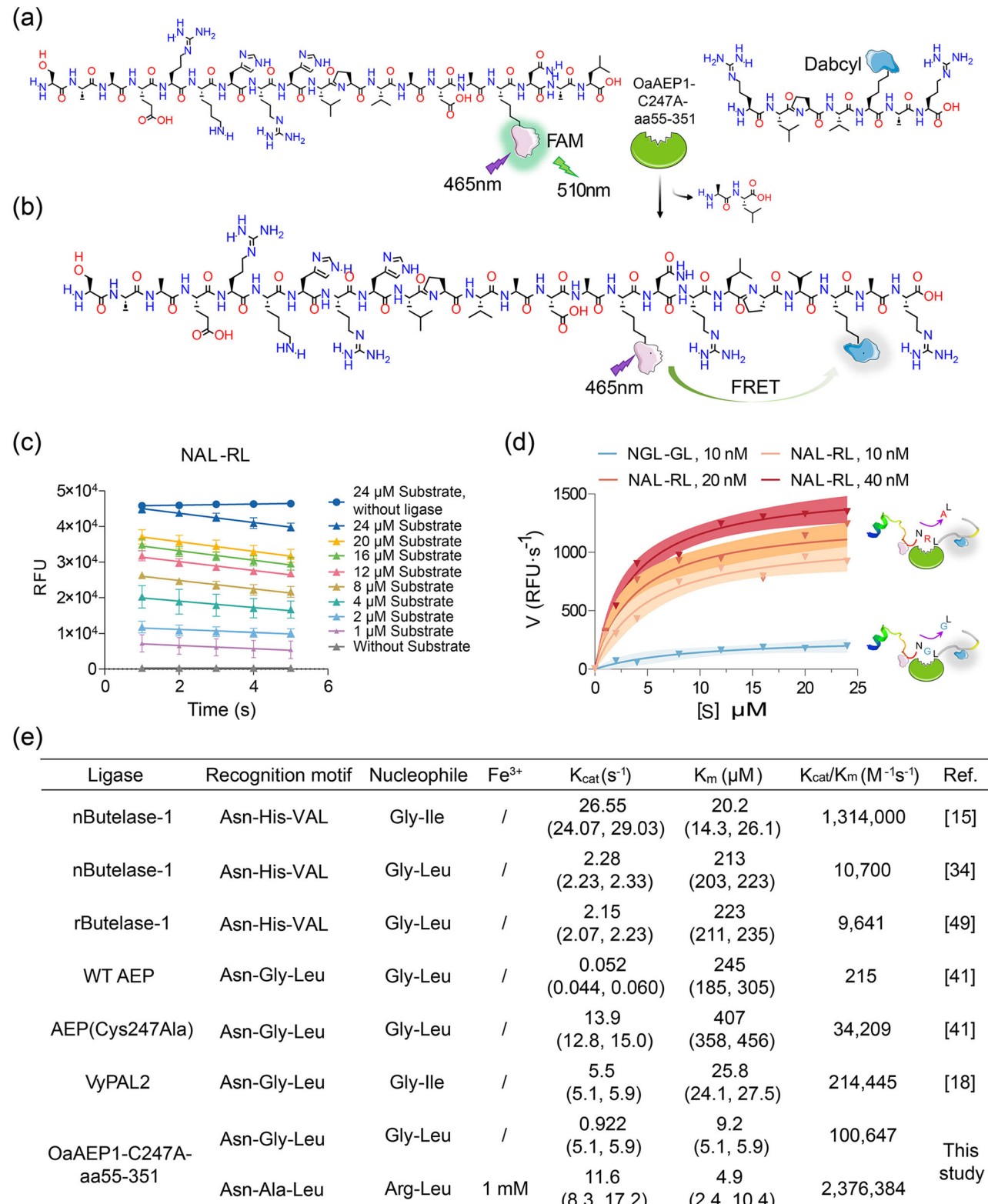

(c) NAL-RL

Legend:
- 24 µM Substrate, without ligase
- 24 µM Substrate
- 20 µM Substrate
- 16 µM Substrate
- 12 µM Substrate
- 8 µM Substrate
- 4 µM Substrate
- 2 µM Substrate
- 1 µM Substrate
- Without Substrate

(d) Legend:
- NGL-GL, 10 nM
- NAL-RL, 20 nM
- NAL-RL, 10 nM
- NAL-RL, 40 nM

(e)

| Ligase | Recognition motif | Nucleophile | $Fe^{3+}$ | $K_{cat}(s^{-1})$ | $K_m$ (µM) | $K_{cat}/K_m$ ($M^{-1}s^{-1}$) | Ref. |
|---|---|---|---|---|---|---|---|
| nButelase-1 | Asn-His-VAL | Gly-Ile | / | 26.55 (24.07, 29.03) | 20.2 (14.3, 26.1) | 1,314,000 | [15] |
| nButelase-1 | Asn-His-VAL | Gly-Leu | / | 2.28 (2.23, 2.33) | 213 (203, 223) | 10,700 | [34] |
| rButelase-1 | Asn-His-VAL | Gly-Leu | / | 2.15 (2.07, 2.23) | 223 (211, 235) | 9,641 | [49] |
| WT AEP | Asn-Gly-Leu | Gly-Leu | / | 0.052 (0.044, 0.060) | 245 (185, 305) | 215 | [41] |
| AEP(Cys247Ala) | Asn-Gly-Leu | Gly-Leu | / | 13.9 (12.8, 15.0) | 407 (358, 456) | 34,209 | [41] |
| VyPAL2 | Asn-Gly-Leu | Gly-Ile | / | 5.5 (5.1, 5.9) | 25.8 (24.1, 27.5) | 214,445 | [18] |
| OaAEP1-C247A-aa55-351 | Asn-Gly-Leu | Gly-Leu | / | 0.922 (5.1, 5.9) | 9.2 (5.1, 5.9) | 100,647 | This study |
| | Asn-Ala-Leu | Arg-Leu | 1 mM | 11.6 (8.3, 17.2) | 4.9 (2.4, 10.4) | 2,376,384 | |

than the ultrafast variant AEP(Cys247Ala), as well as 2 times higher than the reported fastest butelase-1 toward "Asn-His-Val" and "Gly-Ile". The OaAEP1-C247A-aa55-351 holds great promise for a wide range of biotechnological applications. These include in vitro applications, such as protein conjugation, polymerization, and site-specific labeling to functionalized cargo. OaAEP1-C247A-aa55-351 also shows potential in producing bispecific antibodies, as well as peptide or protein cyclization to enhance

stability. Additionally, in planta applications for molecular farming of cyclic therapeutics or pesticides.

## Methods
### Expression and purification of protein

**Ligase**. The gene encoding OaAEP1 was codon-optimized using the Codon Adaptation Tool (http://www.jcat.de/). The nucleotide sequence

**Fig. 5 | Kinetic characteristics of OaAEP1-C247A-aa55–351 in catalyzing the ligation of substrate recognition motif "Asn-Ala-Leu" and nucleophiles "Arg-Leu". a, b** Schematic representation strategy of kinetic study of OaAEP1-C247A-aa55-351 with substrate and nucleophile containing a FRET donor and acceptor, FAM and Dabcyl. Once ligation, Dabcyl comes in proximity with FAM and quenches the fluorescence of FAM emission. **c** Analysis of ligase activity by monitoring the RFU (FAM) value of different concentrations of substrate ("NAL-RL") over time with (40 nM) or without OaAEP1-C247A-aa55–351 using a FRET assay. The ligation reactions were performed with a substrate: nucleophile molar ratio of 1:3, 1 mM $Fe^{3+}$ at 37 °C for 2 min. The decline of RFU of FAM at the first 5 s is used to determine the initial velocity. The assays are performed as $n = 3$ biologically independent experiments. The data are presented as the mean, and the error bars indicate the SD. Simple linear regression was calculated and plotted in GraphPad Prism version 9.0. **d** The variation of catalytic rate of OaAEP1-C247A-aa55-351 with substrate concentration. The catalytic efficiency was calculated with the decreasing rate in fluorescence signal during the first 5 s after enzyme addition. The blue and red curve represents the variation of catalytic rate of OaAEP1-C247A-aa55-351 to "NGL-GL" and new substrate and nucleophile partner "NAL-RL" with 1 mM $Fe^{3+}$, respectively. The shadow represents the error range. **e** Kinetic parameters for the ligation of substrates by OaAEP1-C247A-aa55-351 versus reported WT OaAEP1, Cys247Ala[40], VyPAL2[17], and butelase-1 freshly extracted from a plant (nButelase-1)[14,33] or recombinant expressed (rButelase-1)[48].

is shown in Supplementary data. Considering that aa24-54 and aa326-351 may play essential roles in the activity of ligase, so the gene of aa24-351, aa55-351, aa55-325, aa24-325 was synthesized and cloned into pET32a vector with *Nco* I and *Xho* I as enzyme restriction sites, respectively. The encoding protein is fused with an N-terminal TrxA tag (12 kDa) and a hexa-His tag. Then the plasmids were transformed into *E. coli* BL21 (DE3, maintained in our laboratory), and when cultures grow to exponential growth phase (OD600 of 0.6–0.8), IPTG was added to a final concentration of 0.4 mM to induce recombinant protein expression. The bacteria were further incubated at 16 °C for 6 h. Then, the bacteria were harvested by centrifugation at 7000×*g* for 15 min at 4 °C and resuspended by the lysis buffer containing 50 mM Tris–HCl buffer, pH7.4, 150 mM NaCl, 0.05% (w/v) CHAPS, 10% (v/v) glycerol. The bacteria then were lysed by sonication at 4 °C for 4 min per 500 mL cultures (2 s + 4 s break). The supernatant fraction was separated by centrifugation at 12,000 rpm $min^{-1}$ for 15 min at 4 °C and then purified by cobalt-based IMAC column (Cytiva, NJ, USA). Bound proteins were eluted with a discontinuous imidazole gradient (25, 50, 150, and 250 mM). Protein concentrations were quantified by measuring the absorbance at 280 nm on Multiskan® GO spectrophotometer (Thermo Fisher Scientific, Shanghai, China). The eluted proteins were preserved in 50% glycerol at −20 °C for short-term and at −80 °C for long-term storage.

**rtNP-NAL**. The DNA sequence of truncated nucleocapsid protein (aa1-258) (tNP) of SARS-CoV-2 with C-terminal Asn-Ala-Leu, was inserted into the pET-28a vector with BamH I and Hind III as enzyme restriction sites and expressed in *E. coli*, BL21 (DE3). 0.4 mM IPTG was used to induce rtNP-NAL expression at 25 °C for 6 h. The lysis buffer is 50 mM Tris–HCl buffer, pH8.0 with 50 mM NaCl, and purified by HiTrap SP HP cation exchange chromatography column. Bound proteins were eluted with a discontinuous NaCl gradient (100, 200, 300, 400 mM). The eluted proteins were dialyzed against 50 mM Tris–HCl buffer, pH 8.0.

### Peptides and antibodies
All the peptides (Supplementary Table 1) used in this manuscript were synthesized chemically and purified by HPLC in Sangon (Sangon, Shanghai, China). The HPLC and MS traces of all used peptides is shown in supplementary Data 1. The peptide used as substrate contains a recognition motif (P1–P1'–P2', e.g., Asn-Gly-Leu) of OaAEP1 ligase at the C-terminal of aa133-147 peptide (Pep133) derived from tegument protein pp65 of human cytomegalovirus (HCMV). Nucleophiles are biotinylated or dabcyl-labeled peptides bearing an N-terminal dipeptide nucleophile sequence (P1"–P2", e.g., Gly-Leu). The purity of the synthesized peptides was >90%. The antibody 5D6 against Pep133 and 17H11 against SARS-CoV-2 nucleocapsid protein was prepared in-house using mouse hybridoma technology[55], in which the Balb/c mice were immunized with keyhole limpet hemocyanin-coupled Pep133 or SARS-CoV-2 nucleocapsid protein.

### Ligation assay of truncated OaAEP1
The ligation reaction was conducted in a 50-μL mixture containing buffer with different pH, P1–P1'–P2'-based peptides/proteins, P1"–P2"-based peptides, and protein ligases with or without metal ions. The reaction was performed at 37 °C in a water bath for 30 min.

### Enzyme-linked immunosorbent assay
The ligating products were measured by enzyme-linked immunosorbent assay based on antibodies against the P1–P1'–P2'-based peptides/proteins. Firstly, the microplates were coated by pep133-specific monoclonal antibody 5D6 or SARS-CoV-2 nucleocapsid protein-specific monoclonal antibody 17H11 and used to capture the ligating products at 37 °C for 30 min. After washing the plate five times with PBST (containing 0.05% tween-20), streptavidin-horseradish peroxidase (working dilution: 1:5000, Thermo Fisher Scientific, MA, USA) was added and incubated for 30 min at 37 °C. Then TMB was added for color development after another washing step. The optical density at 450 nm with reference wavelength at 630 nm was determined by a microplate spectrophotometer (Autobio, Zhengzhou, China). All the experiments were performed in triplicate, and the mean ± SD of the three repeats were presented. Statistical analyses were performed using GraphPad Prism version 9.0 (GraphPad Software, San Diego, USA). Multiple groups were compared using ordinary one-way ANOVA.

### Kinetics assay
The kinetic properties of OaAEP1-C247A-aa55-351 in peptide ligation were studied using a Förster resonance energy transfer (FRET) assay. Various concentrations of Pep133-NXL (X = Ala or Gly, FAM labeled) and XL-(X = Arg or Gly, Dabcyl labeled) peptide were mixed at a molar ratio of 1:3 in a 50-μL reaction mixture with 10, 20, 40 nM OaAEP1-C247A-aa55-351 in 20 mM phosphate buffer, pH 7.0. The final concentrations of FAM substrate were 24, 20, 16, 12, 8, 4, 2, 1, and 0 μM. The reactions were run at 37 °C for 2 min and measured at 1-s intervals at the first 5 s to calculate the initial rates during the linear portion of the progress curve. The fluorescence was measured using a Gentier 96E real-time PCR system (Tianlong, Xi'an, China) with an excitation wavelength of 465 nm and an emission wavelength of 510 nm. The velocities (decreasing rate in fluorescence signal of FAM during the first 5 s after enzyme addition) were input into GraphPad Prism version 9.0 to obtain the Michaelis-Menten curve and the kinetic parameters ($K_{cat}$ and $K_m$).

### OaAEP1-C247A-aa55-351 protein LC/MS analysis
2 μg purified OaAEP1-C247A-aa55-351 was separated by SDS–PAGE, and targeted bands were excised, and subjected to reductive alkylation with 5 mM tris(2-carboxyethyl) phosphine hydrochloride (Thermo Fisher Scientific, MA, USA) and 25 mM chloroacetamide (Thermo Fisher Scientific, MA, USA), followed by in-gel digestion with trypsin (Promega, Beijing, China). Digested peptides were extracted with 50% acetonitrile and 0.1% formic acid. Peptides were desalted using C18 desalting tips (Thermo Fisher Scientific, MA, USA) and dried by SpeedVac concentrator. Then the peptides were re-dissolved in 0.1% formic acid and analyzed on an Orbitrap Eclipse Tribrid mass spectrometer (Thermo Fisher Scientific, MA, USA).

### Reporting summary
Further information on research design is available in the Nature Portfolio Reporting Summary linked to this article.

### Data availability
All data supporting the findings of this research are available within the article and its corresponding supplementary information file. The HPLC

chromatograms and LC/MS spectra of the peptides used in this manuscript are available in Supplementary Data 1. The source data of the figures in the main manuscript is available in Supplementary Data 2. All other data or sources are available from the corresponding author on reasonable request.

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

## Acknowledgements

This work was supported by the National Natural Science Foundation of China [grant numbers 31971369 and 82171816], the medical Research Project of Fujian Provincial Health Commission [grant numbers 2021ZD01006], the Industry–University–Academy Co-operation Program of Xiamen [grant numbers 2022CXY0102] and the Fundamental Research Funds for the Central Universities [grant numbers 20720220006 and 20720220005].

## Author contributions

Jiabao Tang: Experiment design and conduct, data analysis, writing—original draft. Mengling Hao: Experiment conduct, data analysis. Junxian Liu, Yaling Chen, Gulimire Wufuer, Xuejie Zhang, Tingquan Zheng: Experiment conduct. Jie Zhu, Shiyin Zhang: Consult. Mujin Fang: Project management, funding acquisition. Tingdong Li, Shengxiang Ge: Supervision, conceptualization, review & editing, funding acquisition. Jun Zhang, Ningshao Xia: Funding acquisition.

## Competing interests

The authors declare no competing interests.
