## [Peer Review File · Communications Chemistry]

Reviewers' comments:

Reviewer #1 (Remarks to the Author):

Reported in this manuscript, a team of biochemists led by Prof Ge and Prof Li conducted a systematic review of truncated constructs of OaAEP1 (C247A). They have found that by tagging with TrxA tag, and by optimizing the boundary of the catalytic domain, OaAEP1 (C247A) could be obtained directly in its activated form and possessing superior catalytic properties.

Considering much of the biochemical investigations have been done, for this particular construct, and its closely related homologs by various research groups in Australia and Singapore, from 2015 onwards, the novelty of the claims shall be tuned down. In general, researchers do not give a new name to a truncated construct of a well-studied enzyme (OaAEP1 (C247a) in this case), and claim it is a 'new ligase'. Nevertheless, the results reported in this manuscript, if validated, could provide convenience in production and application of OaAEP1 (C247A).

The overall write-up and the methodology of this manuscript requires the following clarifications:

1. In the introduction, the existing landscape of AEP-class/PAL class ligases were significantly under reported/summarized. Much of the findings in this manuscript were extensively explored previously. For example, the substrate preference, the active form boundary, the ionic and pH dependent activity profile (Refs, Prof Liu Chuan-fa's works from NTU, Singapore).
2. When comparing the catalytic activities, different methods were used by different groups, thus the kinetic numbers should not be compared directly. The authors were also suggested to test the ligation activities by using 'protein' substrates and using more diverse detection methods to confirm the ligation outcome. Fluorescent method alone, as reported in this manuscript, might face methodological bias. Furthermore, additional controls (oxi-reductive additive? considering the catalytic residue is a Cys) are needed to confirm the effects of metal ions.
3. Please provide additional data on protein purification methods and, since the protein purity in the supplement data did not match the ones shown in Figure 1. In addition, OaAEP1 (C247A) active form will undergo auto cleavage. Thus, please confirm the C-terminal boundary of the active form enzyme using mass-spec.

Overall, there is much room for the scholarship and rigor of the data to be improved.

Reviewer #2 (Remarks to the Author):

This manuscript by Tang et al. describes a strategy to enhance the catalytic efficiency of an Asx-specific ligase of the PAL family. Specifically, the authors reported their design in constructing a new PAL to improve ligase-mediated peptide and protein ligation through (1) truncating the cap domain of OaAEP1-C247A, (2) investigating the optimal combination of recognition motif and the incoming residues, and (3)

exploring the rate-accelerating metal additives. The end result is that they succeeded to obtain a shortened OaAEP1b-C247A, entitled OaAL2, together with the optimized substrate pair “peptide1-NAL + RL-peptide2”. They also claimed that OAL2 gave 30-fold faster k_{cat} than butelase-1, to become the most efficient natural PAL. These results, if found true, are useful to the ligase field. However, there are serious issues with the data, and which could not reconcile with their results or support their conclusion. Revision with new supporting data is recommended.

Comments

(1) As mentioned by the authors, the active form of OaAEP has its inhibitory cap domain self-cleaved under acidic conditions. This is the common characteristics of the legumains. The truncation site of OaAL2 locates after the flexible linker region, which would not be expected to give conformational change to the active core structure. There is no rational explanation why the activity should be improved.

(2) Direct expression of truncated asparaginyl ligases is not novel. It has been reported in 2019 (10.1021/acs.biochem.9b00263). Indeed, no improvement of catalytic activity was reported in truncated version.

(3) In the FRET-based assay, the fluorescent signal changes are just too flat. It is unlikely to give such a high efficiency as reported by the authors. It is worthwhile for the authors to carefully check their calculations. At a minimum, repeat the experiment in triplicates to include SD values. A recent work in Plant Cell (10.1093/plcell/koac281) Figure 2C, which the authors referred to, gave much clear trend in catalytic difference with different substrate concentrations.

(4) Too many selective use and outdated references on PAL. Consequenr, there are striking mis-use and mis-interpretation of references by authors to justify their work. This is unnecessary, but the failure of the authors to update reference is a more serious matter. (see comments below). At present, there are at least 10 publications on PALs since 2017.

(5) The mechanism and substrate specificity of PALs have described and reported (see New Phytologist 238, 1534-1545, 2023; ACS Catalysis, 10,8825-8834, 2020). The claim for mechanism determination in this manuscript is just not there.

(6) Leu is favored at P2'' (NAL) and P2''' (RL), and which is well established. Asparaginyl ligases have broad substrate tolerance, which has been known since their first discovery. Why is a narrow substrate specific of RL an advantage? It should be a disadvantage because it limits the scope of ligation reactions. Please make convincing arguments and cite literature precedents.

(7) Any substrate preference difference between OaAL2 and OaAEP1b-C247A? At the very beginning, the authors claimed that their work is based on the crystal structure of OaAEP1b, hence SAR is expected to be explained to support a high substrate-driven efficiency. But OaEP1b is nowhere as efficient as butelase 1 or VyPAL. Also, of the 12 unique PAL crystal structures reported, there is generally no

difference in their structures or active sites.

(8) Legumains are not metalloproteins. They have cysteine at their active site, and generally do not take metal ions as co-factors. As such, they are known to be inhibited by several metal ions due to their high affinity towards thiols (10.1016/S0167-4838(02)00209-1). Why Fe³⁺ could enhance ligase efficiency? Please explain it thoroughly because this is probably the only new finding in found true. And it is also likely the rational is misguided.

(9) Line 18: Asparaginyl ligases “are” valuable tools. Please correct grammar mistakes here and elsewhere in the text.

(10) Line 46: reaction mediated by the biotin-ligase occurs at Lys side-chain or Na. It is not the same as the rest. Do not mix them under the same term “ligase”. Ref 21 is Sortase and not transferase. The most highly cited papers of Intein and sortase A are not there. Reference on Subtiligase is completely missing.

(11) Line 48: most AEPs are hydrolase but PALs are not. Butelase-1, OaAEP1b, VyPAL2, McPAL, and 15 others (. are homologs of hydrolytic AEPs with different functionality. Do not mix the concept and confuse readers.

(12) Line 55: substrate recognition profile has been studied for butelase-1 in 2014, which covers “NAL-RL” as well.

(13) Line 65: butelase-1 can be recombinantly expressed. This statement is incorrect. Please refer to (10.1021/acs.biochem.9b00263; 10.1021/acs.jafc.1c01755; 10.1039/d1ra03763c). In fact many butelase-like can be expressed efficiently (see

(14) Line 66 and Line 74, why ref [4] and [3]??

(15) Line 145: the reaction was performed with a molar ratio of ligase:substrate = 1:6:60 for 30 min? It does not agree with the high kinetics.

(16) Line 182: hydrolyzed Asn cannot be used for “ligation”. Rephrase the mechanism.

(17) Line 234: cannot see “rising of catalytic rate” in Figure 5C. The concentration of enzyme used in the kinetic study was 0.2 uM and substrate was only 1-25 uM? Difficult to understand 120 s of reactions to get initial rates? It should finish in 1 second if the turnover rate is really 158 s⁻¹. This experiment is problematic.

Reviewer #3 (Remarks to the Author):

The manuscript by Ge and co-workers describes a rationally designed recombinant asparaginyl ligase

with enhanced ligation activity compared to existing ligases. The enzyme could be obtained via a mild purification process with high yield, overcoming the current issues associated with reduced enzymatic activity due to harsh acid activation and low productivity resulting from complicated operating procedures. Importantly, by utilizing the more effective and efficient recognition motif "Asn-Ala-Leu" and alternative nucleophiles "Arg-Leu", the OaAL2 was reported to show higher catalytic activity than the most efficient butelase-1. While this work represents an interesting development in the field and would be potentially useful for peotein/peptide modification, several key issues have to be addressed before any decision can be made:

1) Protein purification: how the protein concentration is measured exactly? Through extinction coefficient? This will be important information for others to reproduce the results. And what is the yield of the proteins through the current recombinant expression technology?

2) The HPLC and MS traces of all used peptides should be given, and why did the authors not try to characterize the ligation reaction through HPLC, but instead, using the rather tedious ELISA protocol?

3) Is the reaction also reversible? As the authors used 5-fold excess of the nucleophile, can it be further reduced? The authors mention in line 207 that "the newly producing ligating tripeptide "Asn-Arg-Leu" motif of ligating product could be less susceptible to hydrolysis", why?

4) More rigorous kinetic studies should be performed to show the high activity of the enzyme, e.g., at different enzyme loading/concentration, and the reaction time and yield are all important information. Authors should refer to ref. 17 for a more complete characterization.

5) In the manuscript, line 255, the authors claimed that "OaAL2 has a Kcat/Km value 30 and 618 times faster than butelase-1 and AEP(Cys247Ala), respectively.". The "30-times" seems to be inconsistent with the results in Figure 5E and the description in line 267: The Kcat/Km of OaAL2 toward recognition motif "Asn-Ala-Leu" and nucleophiles "Arg-Leu" is 16 times faster than the reported fastest butelase 1 toward "Asn-His-Val" and "Gly-Ile". Line 234, "The Kcat value of OaAL2 is 158.4 s⁻¹, approximately 17 times faster than butelase-1[17], the fastest ligase reported till now (Figure 5D-E)." The "17 times" is also inconsistent with the results shown in Figure 5E.

6) It is interesting that Fe³⁺ facilitates the ligating activity of OaAL2 significantly. Were all ligations in this work carried out in the buffer containing Fe³⁺? If so, the concentration should be stated; if not, why? And maybe the effect of Fe³⁺ in different concentrations could be explored to further optimize the ligation activity?

Below are some other minor issues:

1) The English of the manuscript is very poor and should be checked thoroughly? E.g., Line 29 "rational designed" should be "rationally designed"; Line 80, "sharpening" is not the correct word here...

2) The phrase "butelase 1" is not consistent, sometimes as "butelase1," or "butelase-1"

3) Line 126 "Nucleophiles" should be "nucleophiles".

Line to line responses to the reviewer's comments

Dear reviewers:

Thanks for your professional handling and giving us the opportunity to revise our manuscript for further consideration. We have considered the comments carefully and revised the manuscript according to those comments. The comments significantly improved the readability of our manuscript. For your convenience, the revision was done in Track Change mode, and a clean version was also submitted. The comments are laid out below and specific concerns have been numbered. Detailed line-to-line response is given, in which the page and line numbers refers to those in the track change version. Our response is given in **BLUE** normal font.

Reviewer #1 (Remarks to the Author):

Reported in this manuscript, a team of biochemists led by Prof Ge and Prof Li conducted a systematic review of truncated constructs of OaAEP1 (C247A). They have found that by tagging with TrxA tag, and by optimizing the boundary of the catalytic domain, OaAEP1 (C247A) could be obtained directly in its activated form and possessing superior catalytic properties.

Considering much of the biochemical investigations have been done, for this particular construct, and its closely related homologs by various research groups in Australia and Singapore, from 2015 onwards, the novelty of the claims shall be tuned down. In general, researchers do not give a new name to a truncated construct of a well-studied enzyme (OaAEP1 (C247a) in this case), and claim it is a 'new ligase'. Nevertheless, the results reported in this manuscript, if validated, could provide convenience in production and application of OaAEP1 (C247A).

Re: Thank you very much for your comments. Indeed, PALs have been extensively studied by various research groups and their value in various applications have been demonstrated. However, the preparation processes for PALs remain tedious and poorly reproducible, and their activity is unstable, presenting unsolved challenges. Thus, our study aims to gain valuable insights into the key factors that enhance ligating efficiency, ultimately promoting its broad application. Specifically, we focus on OaAEP1-C247A, an asparaginyl endopeptidase capable of joining peptidyl substrates' N- and C-termini upon activation in acidic pH conditions to release the cap. To differentiate it from the zymogen form, we have slightly modified the name by replacing "endopeptidase" (EP)

with “ligase” (L), and the “2” denotes the number of the four recombinant ligases investigated in this study. Following your suggestion, we have removed the term “new” from the Conclusion section of our manuscript.

Line 385-388: “Herein, we explored a rational design approach on the basis of the recently determined crystal structure of the OaAEP1-C247A catalytic domain and constructed a ligase, OaAL2, which eliminates the complicated processes of acid activation and further purification.”

The overall write-up and the methodology of this manuscript requires the following clarifications:

1. In the introduction, the existing landscape of AEP-class/PAL class ligases were significantly under reported/summarized. Much of the findings in this manuscript were extensively explored previously. For example, the substrate preference, the active form boundary, the ionic and pH dependent activity profile (Refs, Prof Liu Chuan-fa's works from NTU, Singapore).

Re: Thank you very much for your professional comments. Indeed, previous research has reported on the substrate preference, the active form boundary and pH dependent activity profile of AEP-class/PAL class ligases. We regret that, due to the initial focus of the original manuscript not being on the optimization of these characteristics, we did not provide a comprehensive summary in the introduction but rather described these aspects within each section of the results. In the revised manuscript, we have provided

a more comprehensive summary of these characteristics of the AEP-class/PAL enzymes in the introduction.

Line 64-76: “Sortase A (srtA) harbors an N-terminal signal peptide, a membrane anchor motif and a core catalytically active domain [42]. LPXTG motif is the recognition sequence of srtA and allows site-specific conjugation. However, its application is limited by the relatively long recognition sequence (LPXTG) and poor the ligating efficiency due to reversible reaction. PALs are expressed as inactive zymogens, contain vacuole-targeted signal peptide, N-terminal pro-domain, catalytic core domain and C-terminal cap domain that covers the active site [43]. Auto-activation occurred at acidic conditions and led to the cleavage of both the pro- and cap domains at the N- and C-termini of the catalytic core and the release of mature active enzyme [35, 36, 38, 41, 44]. The active PALs catalyze transpeptidation at the Asn residue of a short Asn-Xaa1-Xaa2 tripeptide motif at around neutral pH. PALs address the challenge of long recognition sequence seen in Sortase A [34].”

Though previous studies have explored the substrate preference, the active form boundary and pH dependent activity profile of AEP-class/PAL class ligases, **it seems that some problems still remained unsolved:**

(1) **Substrate preference:** AEP-class/PAL class ligases possess the ability to recognize specific sequences of amino acids, and the recognition and nucleophile peptide profiles differ among different ligases. Previous studies have demonstrated the recognized and nucleophilic motif for peptide ligation by OaAEP1. But the results were not so consistent and adequate (PMID: 28369872; 32874509). Moreover, not

all kinds of amino acids were tested in the previous studies, and cross-paired evaluation of the recognition and nucleophilic peptide was absent. Besides, it is unknown for us whether the recognition and nucleophile profiles of OaAL2 are consistent with OaAEP1-C247A. So, in this study, we validated and augmented them. We found that the partner “Asn-Ala-Leu” and “Arg-Leu” reported in this study will enhance the ligation efficiency of OaAL2, thus broad the application of OaAEP1-C247A. We have added associated description in the revised manuscript:

Line 244-246: “Previous studies have demonstrated the recognized and nucleophilic motif for peptide ligation by OaAEP1, but the results were not so consistent and adequate [55, 56].”

(2) **The active form boundary:** indeed, previous studies have reported the boundary of active core domain of PALs. But this active form is achieved by self-cleavage under acidic conditions (PMID: 35480425; 28199119; 36099055). It is currently unknown whether the sequences adjacent to the cleavage site affect enzymatic activity. Moreover, in our study, a decline in ligase activity during acidic activation was observed (Figure R1). Besides, the tedious and poorly reproducible ligases preparation processes and unstable activity of AEP-class/PAL class ligases in the previous study also impede its broad application. To address these issues, a one-step expression and purification process was employed to directly obtain ligases without the complicated processes of acid activation and further purification. Therefore, the negative impact on activity caused by acidic activation can be circumvented. This approach offers convenience in production and application of

OaAEP1 (C247A).

Figure R1. The ligation activity of OaAL2 treated with or without acidic buffer (1 mM EDTA and 0.5 mM TCEP under Acidic buffer (pH 3.7)) to simulate the acidic activation.

(3) **The ionic and pH dependent activity profile:** previous studies have demonstrated that the active form of PALs activated by acidic conditions exhibit ligation activity at around neutral pH. However, the PAL examined in this study, OaAL2, purified without acid activation, raising the question of whether its pH preference profile has been altered. Moreover, previous studies didn't explore the effect of metal ions in ligation efficiency of PALs extensively. Therefore, we aim to conduct a more comprehensive analysis. This study reveals that the pH preference profile of OaAL2 remains unchanged; furthermore, it demonstrates that **metal ions such as Fe³⁺, Zn²⁺ and Mn²⁺ can significantly enhance catalytic efficiency, with Fe³⁺ being the most effective**. **This finding is novel**, given the absence of previous research reports on this topic. Prior studies used “Asn-Gly-Leu-His” as the recognition motif, where divalent cations (Ni²⁺) bind the releasing competing nucleophiles “Gly-Leu-His” in a tetradentate manner via the N-terminal amine, histidyl imidazole, and the

intervening two backbone amides, thus realizing nucleophile quenching via metal complexation to promote the product yields (PMID: 33202079). However, in this study, nucleophile quenching via metal complexation seems infeasible when “Asn-Ala-Leu” and “Arg-Leu” were used as recognition motif and nucleophiles due to the absence of histidyl imidazole. Certainly, combining the strategy of this previous study with our findings, namely using “Asn-Ala-Leu-His” and “Arg-Leu” as recognition motif and nucleophiles, can also be effective with Ni²⁺.

Therefore, we believe that this study will bring new and promising application value to site-specific modification and protein engineering based on chemoenzymatic ligation.

2. When comparing the catalytic activities, different methods were used by different groups, thus the kinetic numbers should not be compared directly. The authors were also suggested to test the ligation activities by using 'protein' substrates and using more diverse detection methods to confirm the ligation outcome. Fluorescent method alone, as reported in this manuscript, might face methodological bias. Furthermore, additional controls (oxi-reductive additive? considering the catalytic residue is a Cys) are needed to confirm the effects of metal ions.

Re: Thank you very much for your professional comments.

(1) About comparing the kinetic numbers directly:

Indeed, comparing the catalytic efficiency of enzymes can be more accurately assessed simultaneously by utilizing the same method. In our attempt to obtain ligase from recombinant proenzyme His-Ub-OaAEP1 as reported in previous study, the protein was

successfully expressed, and its molecular weight was consistent with that reported in the previous study (Figure R2A). However, the protein precipitated during acid activation for unknown reason (Figure R2B). Consequently, it is not possible for us to compare the ligase activity under the same conditions. However, we noticed that some previous reports have used K_{cat}/K_m to compare the catalytic efficiency of different enzymes from different groups (PMID: 33141031; 32221295; 31187974). Consequently, we adopted their presentation mode to relatively compare the K_{cat}/K_m of different PALs. The K_{cat}/K_m of specific substrate to enzyme kinetic is a specificity constant, which is a useful index for comparing the catalytic efficiency of an enzyme acting on alternative substrates. In theory, it is a constant value that does not change with different evaluation methods.

Figure R2. SDS-PAGE analysis of fractions after purification of proenzyme His-Ub-OaAEP1 by Ni-NTA column affinity chromatography (A) and the purified protein before and after acidic activation (B).

(2) Testing the ligation activities by using 'protein' substrates and using more diverse detection methods to confirm the ligation efficiency:

As your suggestions, we utilized SARS-CoV-2 nucleocapsid protein as a validation model to verified that not only peptides, but also properly-folded recombinant protein can be labeled successfully at a specific site under OaAL2 ligation. The DNA sequence of truncated nucleocapsid protein (aa1-258) (tNP) of SARS-CoV-2 with C-terminal Asn-Ala-Leu, was inserted into the pET-28a vector and expressed in *E. coli*, *BL21 (DE3)*. Subsequently, the protein was purified using a HiTrap SP HP cation exchange chromatography column. The recombinant truncated nucleocapsid protein (aa1-258) is designated as rtNP-NAL (Figure R3, Supplementary Figure 3C in the revised supplementary materials). The results show that the biotin labeled peptide can be conjugated to rtNP-NAL, demonstrating that properly-folded recombinant protein is equally capable of being site-specific modified with the OaAL2 (Figure R4, Supplementary Figure 3D in the revised supplementary materials).

Figure R3. SDS-PAGE analysis of fractions after purification of the rtNP-NAL by

HiTrap SP HP cation exchange chromatography column. Super.: supernatant of bacteria lysates after sonication and centrifugation; F.T.: The flow-through fractions of column purification. Elution-1~4 represent the elution fractions of 100, 200, 300, 400 mM NaCl.

Figure R4. The ligation activity of OaAL2 in ligating properly-folded protein substrates. 2 μ M rtNP-NAL and 5 μ M, 10 μ M, 20 μ M RL-(biotin labeled) peptide were ligated by 0.5 μ M OaAL2 at 37°C for 30 min, respectively. The product was detected by ELISA with rtNP antibody, 17H11.

These results are added and discussed in the revised manuscript. Line 344-353:

“Protein labeling is a powerful tool in the field of biomedical and biotechnology. To assess the ability of OaAL2 to facilitate well-folded protein labeling, we expressed and purified recombinant truncated nucleocapsid protein (aa1-258) of severe acute respiratory syndrome coronavirus 2 (SARS-CoV-2) with a C-terminal “Asn-Ala-Leu”, designated as rtNP-NAL (Supplementary Figure 3C). Nucleophiles in the form of “Arg-Leu” (RL- (biotin labeled) peptide) were used. The resulting product was detected using ELISA similar to Figure 2A, utilizing the antibody against rtNP, 17H11 coated on the microplate. The results demonstrated that the biotin labeled peptide could be conjugated to rtNP-NAL (Supplementary Figure 3D), indicating that properly-folded recombinant proteins are equally capable of site-specific modification with the assistance of OaAL2.”

In response to your concern regarding methodological bias, the ligation efficiency was

characterized using SDS-PAGE. To facilitate clear differentiation between the ligation products and the substrates before ligation, 50 amino acids long peptide C50-NAL and RL- (biotin labeled) polyXXK peptide were ligated under OaAL2. The results showed that more than 90% of C50-NAL was successfully ligated with RL- (biotin labeled) polyXXK peptide (Figure R5), providing evidence that OaAL2 can achieve high yields through effective and efficient recognition and alternative nucleophiles.

Figure R5. SDS-PAGE analysis of the yield of product after ligating by OaAL2. 5 μM C50-NAL and 50 μM RL- (biotin labeled) polyXXK peptide were ligated by 1 μM OaAL2 and 1mM Fe^{3+} in phosphate buffer, pH 7.2 at 37°C for 30 min and 1 hour, respectively. The ligating product were subjected to analyzed by SDS-PAGE. The bands of OaAL2, ligation product, C50-NAL and GL- (biotin labeled) polyXXK peptide were annotated in the figure.

These results are added and discussed in the revised manuscript. Line 337-343:

“To Further assess the product yield following ligation by OaAL2, we synthesized 50 amino acids long peptide C50-NAL and RL- (biotin labeled) polyXXK peptide to

observe noticeable changes between the ligation products and the substrates prior to ligation. After ligation by OaAL2 under the buffer with Fe³⁺, the products were subjected to analysis by SDS-PAGE. The results indicated that over 90% of C50-NAL was ligated with RL- (biotin labeled) polyXXK peptide (Supplementary Figure 3A-B), demonstrating that OaAL2 is capable of producing high yields of products.”

(3) Validating the Oxi-reductive effects of metal ions on increasing the ligase activity:

As the suggestions of the reviewer, considering the catalytic residue is a Cys, we included additional oxi-reductive additive (oxidizing agent: H₂O₂; reducing agent: β-mercaptoethanol, β-ME) to examine the oxidation-reduction effects of metal ions on increasing the ligase activity. The commonly used concentration of H₂O₂ as oxidizing agent is 10-200 mM (0.03%-0.6%) (PMID: 29695131; 30250002), β-ME as reducing agent is 0.1%-5% (PMID: 22767388; 22947094; 33114565; 36437524; 21855885; the most commonly used is 0.1%). However, it appears that both the oxidizing agent (H₂O₂) and the reducing agent (β-mercaptoethanol) had an adverse effect on the ligation activity of OaAL2, and the former seemed worse (Figure R6). This suggests that metal ions may have other functions that play a significant role in promoting the ligation reaction. In fact, the current research findings are insufficient to provide a comprehensive interpretation for this issue, leaving it as an exciting subject for future investigations in our lab.

Figure R6. The effect of oxi-reductive additive in promoting the ligating efficiency of OaAL2. 2 μ M Pep133-NAL and 10 μ M RL-(biotin labeled) peptide were ligated by 0.2 μ M OaAL2 with different concentrations of β -mercaptoethanol (β -ME, reducing agent) or H₂O₂ (Oxidizing agent) at 37°C for 30 min.

These results are added and discussed in the revised manuscript. Line 286-292:

“Given that the catalytic residue is cystine, we introduced oxi-reductive additives to examine the oxidation-reduction effects of metal ions on enhancing the ligase activity. However, both the oxidizing agent (H₂O₂) and the reducing agent (β -mercaptoethanol) appeared to have detrimental effect on the ligation activity of OaAL2, and the former seemed worse (Supplementary Figure 2D). Therefore, the role of metal ions necessitates further exploration for a comprehensive understanding in future studies.”

3. Please provide additional data on protein purification methods and, since the protein purity in the supplement data did not match the ones shown in Figure 1. In addition, OaAEP1 (C247A) active form will undergo auto cleavage. Thus, please confirm the C-terminal boundary of the active form enzyme using mass-spec.

Re: Thank you very much for your comments. We did not have any other purification

steps apart from those described in the manuscript. The mismatch in protein purity between Figure 1 and Figure S1 could be due to the difference in the loading quantity of the samples. We have adjusted the loading quantity and re-electrophoresed the samples (Figure R7), and the updated result is presented in the revised supplemental data (Supplementary Figure 1C).

Figure R7. SDS-PAGE analysis of fractions after purification of the truncated ligases by cobalt-based IMAC column. Elution-No. was represented fractions eluted with 25, 50, 150 and 250 mM imidazole, respectively.

Only the core domain (ligase without cap domain) was expressed in this study, so it should not undergo auto cleavage, theoretically. To determine the C-terminal boundary of OaAL2, amino acid sequencing was performed by LC-MS/MS. The amino acid sequence of the recombinant OaAL2 served as a template for obtaining the sequence.

Auto cleavage was not observed from the result of LC-MS/MS. The amino acid sequence obtained was consistent with the theoretical sequence (Figure R8, Figure 2D in the revised manuscript).

Figure R8. Sequence coverage view of OaAL2 protein by LC-MS/MS analysis. The amino acids are colored in red, blue and black, which represents three different levels of amino acid confidence from high to low.

These results are added and discussed in the revised manuscript. Line 176-183:

“To determine whether auto-cleavage occurred during the expression and purification of OaAL2, amino acid sequencing was performed to determine its C-terminal boundary. Purified OaAL2 proteins were separated by SDS-PAGE, followed by in-gel digestion with trypsin, and subjected to amino acid sequencing by LC-MS/MS. The theoretical amino acid sequence of the recombinant OaAL2 (Supplementary Figure 1A) was used as a template to obtain the sequence. Auto cleavage was not observed according to the result of LC-MS/MS. The amino acid sequence was consistent with the theoretical sequence (Figure 2D).”

Reviewer #2 (Remarks to the Author):

This manuscript by Tang et al. describes a strategy to enhance the catalytic efficiency of an Asx-specific ligase of the PAL family. Specifically, the authors reported their design in constructing a new PAL to improve ligase-mediated peptide and protein ligation through (1) truncating the cap domain of OaAEP1-C247A, (2) investigating the optimal combination of recognition motif and the incoming residues, and (3) exploring the rate-accelerating metal additives. The end result is that they succeeded to obtain a shortened OaAEP1b-C247A, entitled OaAL2, together with the optimized substrate pair “peptide1-NAL + RL-peptide2”. They also claimed that OAL2 gave 30-fold faster k_{cat} than butelase-1, to become the most efficient natural PAL. These results, if found true, are useful to the ligase field. However, there are serious issues with the data, and which could not reconcile with their results or support their conclusion. Revision with new supporting data is recommended.

Comments

(1) As mentioned by the authors, the active form of OaAEP has its inhibitory cap domain self-cleaved under acidic conditions. This is the common characteristics of the legumains. The truncation site of OaAL2 locates after the flexible linker region, which would not be expected to give conformational change to the active core structure. There is no rational explanation why the activity should be improved.

Re: Thank you very much for your professional comments. We agree with your opinion that removing the cap domain does not theoretically alter the conformation of the active

core structure, and therefore should not enhance the activity. The main factors contributing to the enhanced activity of OaAL2, as demonstrated in this study, are as follows:

- a) **Purification conditions:** no acidic activation is required. In the purification process of OaAEP1-C247A, acidic activation (pH 3.4-4.0) is necessary for OaAEP1-C247A to facilitate self-cleavage and release of the cap domain. To simulate the acidic activation, we treated OaAL2 with 1 mM EDTA and 0.5 mM TCEP under Acidic buffer (pH3.7). We observed a decline in ligase activity during this process (Figure R1, Page 6). Importantly, acidic self-activation is not required in our study. Therefore, the negative impact on activity caused by acidic activation can be circumvented. This is the primary reason for the expected improvement in activity.
- b) **Recognition and nucleophilic motifs:** In the previous study, recognition motifs “Asn-Gly-Leu” and nucleophiles “Gly-Leu” was used. However, our study verified that they exhibit lower ligation efficiency compared to "Asn-Ala-Leu" and an N-terminus "Arg-Leu" (Figure 4E in the revised manuscript). The latter combination significantly enhances the product yields, which play a crucial role in improving ligating efficiency.
- c) **ligating conditions:** Our study also found that the presence of Fe³⁺ enhances the ligating efficiency of OaAL2, further contributing to the overall improvement.

As mentioned above, we believe that all three factors should ultimately contribute to the enhancement of OaAL2 in terms of ligating efficiency.

(2) Direct expression of truncated asparaginyl ligases is not novel. It has been reported in 2019 (10.1021/acs.biochem.9b00263). Indeed, no improvement of catalytic activity was reported in truncated version.

Re: Thank you very much for your comments. We appreciate your attention to previously published studies in the field. While it is true that Pi et al. directly expressed butelase-1 successfully via yeast, there are several important distinctions that set our study apart and contribute to its novelty.

(1) Direct expression of OaAEP1-C247A hasn't been reported and the expression and purification processes in this study are simple and time-saving. The directly expression processes of butelase-1 seem complicated (multiple rounds of pre-cultivation steps are required) and time-consuming (4-5 days, adding methanol every day, and purification processes are not included). While in this study, expression processes are easy and only take 1 day.

Indeed, in order to compare the catalytic activity of the truncated version (OaAL2 in this study) with the proenzyme after activation, we attempted to obtain ligase from the recombinant proenzyme His-Ub-OaAEP1. The gene was inserted into the coding region (NcoI-NdeI) of the pET-28b (+) vector, and recombinant expression in *E. coli* *BL21 (DE3)* was carried out following the previously reported protocol. The protein was successfully expressed, and its molecular weight was consistent with that reported in the previous study (Figure R2A, page 8). However, unexpectedly, the protein precipitated during acid activation (Figure R2B, page 8). Consequently, it is a hopeless

attempt for us to compare the ligase activity from acid activation and direct expression of truncated ligases under the same condition.

(2) More effective recognition and nucleophilic motifs will enhance the broad application of OaAEP1-C247A. As the previous studies have produced inconsistent and inadequate results about the efficient recognition and nucleophile peptides of OaAEP1-C247A (PMID: 28369872; 32874509), and it is unknown for us whether the recognition and nucleophile peptide profiles of OaAL2 are consistent with OaAEP1-C247A. So, in this study, we validated and augmented the profiles of recognition and nucleophile peptides. The newly presented partner “Asn-Ala-Leu” and “Arg-Leu” was proved to enhance the ligation efficiency of OaAL2, which will enhance the broad application of OaAEP1-C247A.

(3) Metal ion Fe^{3+} 、 Zn^{2+} and Mn^{2+} could effectively increase catalytic power, and Fe^{3+} performs best. This finding is novel, as there are no previous research reports on this. Prior studies used “Asn-Gly-Leu-His” as the recognition motif, where divalent cations (Ni^{2+}) bind the releasing competing nucleophiles “Gly-Leu-His” in a tetradentate manner via the N-terminal amine, histidyl imidazole, and the intervening two backbone amides, thus realizing nucleophile quenching via metal complexation to promote the product yields (PMID: 33202079). However, in this study, nucleophile quenching via metal complexation seems infeasible when “Asn-Ala-Leu” and “Arg-Leu” were used as recognition motif and nucleophiles due to the absence of histidyl imidazole. Certainly, combining the strategy of this previous study with our findings,

namely using “Asn-Ala-Leu-His” and “Arg-Leu” as recognition motif and nucleophiles, can also be effective with Ni^{2+} .

Therefore, we believe that all these contributed to the novelty of this study, and will bring new and promising application value to site-specific modification and protein engineering based on chemoenzymatic ligation.

(3) In the FRET-based assay, the fluorescent signal changes are just too flat. It is unlikely to give such a high efficiency as reported by the authors. It is worthwhile for the authors to carefully check their calculations. At a minimum, repeat the experiment in triplicates to include SD values. A recent work in *Plant Cell* (10.1093/plcell/koac281) Figure 2C, which the authors referred to, gave much clear trend in catalytic difference with different substrate concentrations.

Re: Thank you very much for your comments. We apologize for our miscalculation of K_{cat} in the initial manuscript, where V_{max} was determined based on the decrease in fluorescent signal of FAM, rather than the molarity of substrate or ligating product. This miscalculation has been rectified. According to your suggestions, we conducted the FRET-based experiment in triplicates to verify the results and the SD value was showed in the figures. Furthermore, in the initial manuscript, the initial reaction was not well-controlled because the ligation is running immediately after the enzyme was added. The first reaction has been running approximately 30-60 s when the last reaction begins. Hence, we optimized the operational steps. Specifically, we separated the substrates and ligases in the tube and the inner side of the tube cap. The tube was then

gently capped, and fluorescence signals of FAM were collected immediately after rapid mixing and transient centrifugation. Besides, to avoid an excessively fast reaction that may hinder the accurate determination of the initial rate, the concentration of OaAL2 is reduced. Meanwhile, in the revised manuscript, we optimized the concentration of Fe^{3+} and found that 1mM was the most suitable concentration to facilitate the yield of ligating product (Figure R9). Based on the optimized conditions mentioned above, we reproduced the FRET-based experiment, compared the ligating efficiency before and after the optimization. The rising of catalytic rate was more prominent (Figure 5C in the revised manuscript). The results showed that the $K_{\text{cat}}/K_{\text{m}}$ of OaAL2 toward recognition motif “Asn-Ala-Leu” and nucleophiles “Arg-Leu” with 1 mM Fe^{3+} is superior to the previously reported “Asn-Gly-Leu” and “Gly-Leu” (Figure R10D-E, Figure 5D-E in the revised manuscript). Moreover, benefiting from the optimizing concentration of Fe^{3+} , K_{m} of OaAL2 toward “Asn-Ala-Leu” and “Arg-Leu” in the revised manuscript is better than that in the initial manuscript (Figure R10E). In addition, we are sorry that the turnover rate was miscalculated in the original manuscript, and we have corrected the calculation in the revised manuscript.

Figure R9. Different concentrations of Fe^{3+} in effecting the ligation activity of OaAL2.

Figure R10. Kinetic characteristics of OaAL2 in catalyzing the ligation of substrate recognition motif “Asn-Ala-Leu” and nucleophiles “Arg-Leu”. (A, B) Schematic representation strategy of kinetic study of OaAL2 with substrate and nucleophile containing a FRET donor and acceptor, FAM and Dabcyl. Once ligation, Dabcyl comes in proximity with FAM and quenching the fluorescence of FAM emission. (C) Analysis of ligase activity by monitoring the RFU (FAM) value of different concentration of substrate (“NAL-RL”) over time with (40 nM) or without OaAL2 using a FRET assay. The ligation reactions were performed with a substrate: nucleophile molar ratio of 1: 3, 1mM Fe³⁺ at 37°C for 2 min. The declining of RFU of FAM at the first 5 seconds is used to determine the initial velocity. Experiment was performed in triplicates. The data are presented as the mean, and the error bars indicate the SD. Simple linear regression was calculated and plotted in GraphPad Prism version 9.0. (D) The variation of catalytic rate of OaAL2 with substrate concentration. The catalytic efficiency was calculated with the decreasing rate in

fluorescence signal during the first 5 s after OaAL2 (10, 20, 40 nM) addition. The blue and red curve represents the variation of catalytic rate of OaAL2 to “NGL-GL” and new substrate and nucleophile partner “NAL-RL” with 1mM Fe³⁺, respectively. The shadow represents the error range. (E) Kinetic parameters for the ligation of substrates by OaAL2 versus reported WT OaAEP1, Cys247Ala (PMID: 28199119), VyPAL2 (PMID: 34096285) and butelase-1 freshly extracted from a plant (nButelase-1) (PMID: 26633100; 25038786) or recombinant expressed (rButelase-1) (PMID: 34003638).

Line 323-336: “The catalytic rate exhibited a rapid rise with increasing substrate concentration, reaching a plateau thereafter (Figure 5D). Notably, the K_{cat}/K_m value of OaAL2 towards the substrate recognition motif “Asn-Ala-Leu” and nucleophiles “Arg-Leu” under the condition of Fe³⁺ was approximately 24 times higher than OaAL2 towards the previously reported “Asn-Gly-Leu” and “Gly-Leu”, and 70 times higher than the ultrafast variant AEP(Cys247Ala), as well as 2 times higher than butelase-1 [15], the most efficient ligase reported to date (Figure 5D-E). Interestingly, OaAL2 exhibited markedly enhanced catalytic activity compared to both butelase-1 and AEP(Cys247Ala). This improvement can primarily be attributed to the mild purification process, which circumvents the reduction in enzyme activity associated with acidic activation, as well as the enhanced ligating yield benefiting from more efficient recognition and nucleophile motifs, as well as the incorporation of metal ions.”

(4) Too many selective use and outdated references on PAL. Consequently, there are striking mis-use and mis-interpretation of references by authors to justify their work. This is unnecessary, but the failure of the authors to update reference is a more serious matter. (see comments below). At present, there are at least 10 publications on PALs since 2017.

Re: Thank you very much for your comments. We are sorry for our careless omission of some references. We have updated reference as your suggestions below (comment 5, 10, 13, 14).

(5) The mechanism and substrate specificity of PALs have described and reported (see New Phytologist 238, 1534-1545, 2023; ACS Catalysis, 10,8825-8834, 2020). The claim for mechanism determination in this manuscript is just not there.

Re: Thank you very much for your comments. Hemu et al. have described and reported that the S2 and S1' substrate-binding pockets of PALs immediately flanking the catalytic S1 site constitute the major ligase activity determinants (LADs), control the catalytic directionality of the enzymes. The C-terminal tripeptide recognition motif is broken between P1 and P1' as the catalytic cysteine of PALs performs a nucleophilic attack and leads to the formation of the acyl-enzyme intermediate between the catalytic cysteine and the P1 Asn residue. The S-acyl intermediate is positioned at the S2 site and the substrate orientated, controlling accessibility of either water or incoming nucleophiles from the prime-side (S1' site) of the catalytic center. We have rephrased the mechanism in Line 236-243:

“Previous studies have reported the C-terminal tripeptide recognition motif of OaAEP1, Asn-Gly-Leu (NGL, P1-P1'-P2') is broken between P1 and P1' during the catalytic process. This occurs as the catalytic cysteine of OaAEP1 performs a nucleophilic attack and leads to the formation of the acyl-enzyme intermediate between the catalytic cysteine and the P1 Asn residue [35, 38, 41, 44, 53, 54]. Then, the amine group of the N-terminal Gly-Leu (GL)-based nucleophile peptide (P1''-P2'') attack onto the formed

unstable acyl-enzyme intermediate breaks the transient thioester bond and releases the ligating product from the catalytic cysteine [38, 44, 53].”

(6) Leu is favored at P2” (NAL) and P2”” (RL), and which is well established. Asparaginyl ligases have broad substrate tolerance, which has been known since their first discovery. Why is a narrow substrate specific of RL an advantage? It should be a disadvantage because it limits the scope of ligation reactions. Please make convincing arguments and cite literature precedents.

Re: Thank you very much for your comments. We agree with you that broad substrate tolerance is an advantage of asparaginyl ligases, which could broaden the scope of application. In fact, we emphasize the higher ligation efficiency of OaAL2 with “NAL-RL”, and this will increase the yield of the ligating product, thus benefit and push adoption further.

(7) Any substrate preference difference between OaAL2 and OaAEP1b-C247A? At the very beginning, the authors claimed that their work is based on the crystal structure of OaAEP1b, hence SAR is expected to be explained to support a high substrate-driven efficiency. But OaAEP1b is nowhere as efficient as butelase 1 or VyPAL. Also, of the 12 unique PAL crystal structures reported, there is generally no difference in their structures or active sites.

Re: Thank you very much for your comments. Indeed, Hemu et al. have reported the mechanism and substrate specificity of PALs and seems that the structures and active sites are similar. Theoretically, the substrate preference of OaAL2 is expected to be similar to that of OaAEP1b-C247A. However, previous studies have produced

inconsistent and inadequate results about the efficient recognition and nucleophile peptides of OaAEP1-C247A (PMID: 28369872; 32874509), and cross-paired evaluation of the recognition and nucleophilic peptide was absent. The objective of this study is to validate and complement its recognizing and nucleophilic profiles, and discovered that “NAL-RL” exhibits higher efficiency compared to the commonly used “NGL-GL”. But due to the unsuccessful purification of activated OaAEP1-C247A (Response to Comment 2), the preference of OaAEP1-C247A to “NAL-RL” could not be validated.

(8) Legumains are not metalloproteins. They have cysteine at their active site, and generally do not take metal ions as co-factors. As such, they are known to be inhibited by several metal ions due to their high affinity towards thiols (10.1016/S0167-4838(02)00209-1). Why Fe³⁺ could enhance ligase efficiency? Please explain it thoroughly because this is probably the only new finding in found true. And it is also likely the rational is misguided.

Re: Thank you very much for your comments. We agree with you that Fe³⁺ could enhance ligase efficiency was an important finding of our study. Yamane et al. reported that the activity of legumain (asparaginyl endopeptidase) was inhibited by Hg²⁺ and Cu²⁺ and Cd²⁺, but the metal ions Fe³⁺, Zn²⁺, Mn²⁺ found effective in this study was not validated. And it was also observed in this study that not all metal ions are capable of enhancing enzymatic catalytic activity. So, there is no contradiction between the two studies. To investigate the reason that Fe³⁺ enhance ligation efficiency, according to the suggestion of the other reviewer of this manuscript, we included additional oxi-

reductive additive to examine the oxidation-reduction effects of metal ions on increasing the ligase activity. However, it appears that both the oxidizing agent (H_2O_2) and the reducing agent (β -mercaptoethanol) had an adverse effect on the ligation activity of OaAL2, and the former seemed worse (Figure R6 in page 13). This suggests that metal ions may have other functions that play a significant role in promoting the ligation reaction (see details in the response to comment 2 of the reviewer#1, Page 12-13). However, we consistently observed and replicated the enhancing effect of Fe^{3+} through multiple repetitions of our experiments, confirming its genuineness and repeatability. The current research findings are insufficient to provide a comprehensive interpretation for this issue, leaving it as an exciting subject for future investigations in our lab. We have deleted the associated description that metal ions can act as cofactor of Legumains in the revised manuscript.

These results are added and discussed in the revised manuscript. Line 282-292:

“Based on the optimal substrate of Asn-Ala-Leu (P1-P1’-P2’) and nucleophile peptide of Arg-Leu (P1’’-P2’’), the effect of Fe^{3+} were further validated and the optimum concentration of Fe^{3+} were investigated. The result demonstrated that a moderate concentration of Fe^{3+} enhances the yield of ligating product, with 1 mM proving to be the most effective (Figure 4F). Given that the catalytic residue is cystine, we introduced oxi-reductive additives to examine the oxidation-reduction effects of metal ions on enhancing the ligase activity. However, both the oxidizing agent (H_2O_2) and the reducing agent (β -mercaptoethanol) appeared to have detrimental effect on the ligation activity of OaAL2, and the former seemed worse (Supplementary Figure 2D).

Therefore, the role of metal ions necessitates further exploration for a comprehensive understanding in future studies.”

(9) Line 18: Asparaginyl ligases “are” valuable tools. Please correct grammar mistakes here and elsewhere in the text.

Re: We are so sorry for our mistakes. We have corrected them in the revised manuscript.

Line 18-19: “Asparaginyl ligases have been extensively utilized as valuable tools for site-specific bioconjugation or surface-modification.”

(10) Line 46: reaction mediated by the biotin-ligase occurs at Lys side-chain or Na. It is not the same as the rest. Do not mix them under the same term “ligase”. Ref 21 is Sortase and not transferase. The most highly cited papers of Intein and sortase A are not there. Reference on Subtiligase is completely missing.

Re: We sincerely thank the reviewer for careful reading. As suggested, we have deleted the biotin-ligase associated reference (ref 15 and 16 of initial manuscript) and transferred the initial ref 21 (ref 19 in the revised manuscript) to sortase. References on highly cited papers of Intein (Ref 23-27), sortase A (Ref 17, 19, 28 and 33) and Subtiligase (Ref 20 and 21) are supplemented in the revised manuscript.

(11) Line 48: most AEPs are hydrolase but PALs are not. Butelase-1, OaAEP1b, VyPAL2, McPAL, and 15 others (are homologs of hydrolytic AEPs with different functionality. Do not mix the concept and confuse readers.

Re: Thank you very much for your comments. We are sorry for our misleading description. We have modified the related description in the revised manuscript.

Line 58-64: “Notably, the peptide/protein ligases currently identified are primarily

composed of Sortase A [17, 19, 28, 33] and peptide asparaginyl ligases (PALs), such as butelase-1 [15, 34], OaAEP1b [35], HeAEP3 [36], OaAEP3-5 [37], VyPAL2 [18, 38], which facilitate the formation of peptide bonds [15, 34, 35, 39-41].”

(12) Line 55: substrate recognition profile has been studied for butelase-1 in 2014, which covers “NAL-RL” as well.

Re: Thank you very much for your comments. In 2014, nguyen et al. reported the discovery of butelase-1, which exhibited a broad specificity for the N-terminal amino acids of peptide substrates. The study demonstrated that a C-terminal HV dipeptide is essential for efficient recognition by butelase-1, with NHV markedly more efficient (PMID: 25038786). Regarding the N-terminal specificity of the acceptor nucleophile, butelase-1 can accommodate most natural amino acids at the P1” position, excluding proline and acidic amino acids like aspartate and glutamate. Additionally, butelase-1 exhibited a more stringent requirement at the P2” position compared to the P1” position, displaying a pronounced preference for hydrophobic amino acids, particularly isoleucine, leucine and valine (PMID: 25038786). It seems that “NAL-RL” found in our study was not covered.

The investigation of substrate recognition profiles plays a vital role in the application of PALs. Our study is aimed at finding the higher efficiency of substrate recognition and nucleophile profile to accelerate its broad application.

(13) Line 65: butelase-1 can be recombinantly expressed. This statement is incorrect.

Please refer to (10.1021/acs.biochem.9b00263; 10.1021/acs.jafc.1c01755; 10.1039/d1ra03763c). In fact, many butelase-like can be expressed efficiently (see

Re: Thank you very much for your comments. We are sorry for overlooking some researches during retrieving literatures. We have modified the related description in the revised manuscript.

Line 93-96: “Recent advances have enabled the recombinant expression of butelase-1, however, it is still challenged by the unsatisfactory yields, undesirable catalytic efficiency and complicated, time-consuming expression and purification processes [47-49].”

(14) Line 66 and Line 74, why ref [4] and [3]??

Re: We sincerely thank the reviewer for careful reading. We extend our sincere apologies for the citation error that occurred. In using EndNote, the correct reference we aimed to insert was positioned adjacent to the one erroneously cited. This proximity caused an oversight in the selection process of references during the importation stage, resulting in an inaccurate citation. We have reviewed and make appropriate changes to the references throughout the manuscript.

Since we have modified the related description in line 66 of the previous manuscript, so associated references have been updated.

Line 93-96: “Recent advances have enabled the recombinant expression of butelase-1, however, it is still challenged by the unsatisfactory yields, undesirable catalytic efficiency and complicated, time-consuming expression and purification processes [47-49].”

Line 104-106: “Full-length OaAEP1 can be expressed in the *E. coli* and is self-activated under acidic conditions (pH 3.4-4.0) to release the cap [41].”

(15) Line 145: the reaction was performed with a molar ratio of ligase: substrate = 1:6:60 for 30 min? It does not agree with the high kinetics.

Re: We appreciate your insightful comments. It is important to note that the molar ratios of ligase to substrate used in Figure 2B and the kinetics study (Figure 5) differ. The reaction detailed in Figure 2B was an initial comparison of ligation activity among four truncated OaAEP1 variants under basic experimental conditions, prior to any optimization. Hence, at this preliminary stage, with unoptimized ligating conditions and substrate profiles, the substrates Pep133-NGL (3 μ M) and GL -(biotin labeled) peptide (30 μ M) were employed with a higher molar ratio of ligase (0.5 μ M) to substrates (1:6:60), and without the inclusion of Fe^{3+} . In contrast, the kinetics study was performed under optimal ligation conditions with a molar ratio of Pep133-NAL to RL-(biotin labeled) peptide of 1:3, including Fe^{3+} , and OaAL2 at a significantly reduced concentration of 10-40 nM, which is substantially lower than that used in Figure 2B. These methodological differences are responsible for the disparities observed in the results.

(16) Line 182: hydrolyzed Asn cannot be used for “ligation”. Rephrase the mechanism.

Re: Thank you very much for your professional comments. We have rephrased the mechanism in Line 236-243 in the revised manuscript.

“Previous studies have reported the C-terminal tripeptide recognition motif of OaAEP1, Asn-Gly-Leu (NGL, P1-P1'-P2') is broken between P1 and P1' during the catalytic process. This occurs as the catalytic cysteine of OaAEP1 performs a nucleophilic attack and leads to the formation of the acyl-enzyme intermediate between the catalytic

cysteine and the P1 Asn residue [35, 38, 41, 44, 53, 54]. Then, the amine group of the N-terminal Gly-Leu (GL)-based nucleophile peptide (P1''-P2'') attack the unstable acyl-enzyme intermediate, which breaks the transient thioester bond and releases the ligating product from the catalytic cysteine [38, 44, 53].”

(17) Line 234: cannot see “rising of catalytic rate” in Figure 5C. The concentration of enzyme used in the kinetic study was 0.2 μM and substrate was only 1-25 μM ? Difficult to understand 120 s of reactions to get initial rates? It should finish in 1 second if the turnover rate is really 158 s^{-1} . This experiment is problematic.

Re: Thank you very much for your comments. In the initial manuscript, the initial reaction was not well-controlled because the ligation is running immediately after the enzyme was added. The first reaction has been running approximately 30-60 s when the last reaction begins. Hence, we optimized the operational steps just as response to comment 3 (Page 20-23). We reconducted the FRET-based experiment, compared the ligating efficiency before and after the optimization. In addition, we are sorry that the turnover rate was miscalculated in the original manuscript, and we have corrected the calculation in the revised manuscript. The turnover rate of OaAL2 towards “Asn-Ala-Leu” and “Arg-Leu” is 11.6 s^{-1} . Please see details in response to comment 3.

Reviewer #3 (Remarks to the Author):

The manuscript by Ge and co-workers describes a rationally designed recombinant asparaginyl ligase with enhanced ligation activity compared to existing ligases. The enzyme could be obtained via a mild purification process with high yield, overcoming the current issues associated with reduced enzymatic activity due to harsh acid activation and low productivity resulting from complicated operating procedures. Importantly, by utilizing the more effective and efficient recognition motif “Asn-Ala-Leu” and alternative nucleophiles “Arg-Leu”, the OaAL2 was reported to show higher catalytic activity than the most efficient butelase-1. While this work represents an interesting development in the field and would be potentially useful for protein/peptide modification, several key issues have to be addressed before any decision can be made:

1) Protein purification: how the protein concentration is measured exactly? Through extinction coefficient? This will be important information for others to reproduce the results. And what is the yield of the proteins through the current recombinant expression technology?

Re: Thank you very much for your comments. The protein concentration is determined by measuring the absorbance at 280 nm on a UV spectrophotometer. We have supplemented the detailed information in the revised manuscript.

The total yield of OaAL2 through the current recombinant expression technology is about 9 mg per gram bacteria, and we found that the activity of proteins eluted with different imidazole gradient (50, 150 and 250 mM) are distinct. The most active portion

(elution at 50 mM imidazole) is approximately 4.2 mg per gram bacteria.

Line 424-426: “Protein concentrations were quantified by measuring the absorbance at 280 nm on Multiskan® GO spectrophotometry (Thermo Fisher Scientific, Shanghai, China).”

2) The HPLC and MS traces of all used peptides should be given, and why did the authors not try to characterize the ligation reaction through HPLC, but instead, using the rather tedious ELISA protocol?

Re: Thank you very much for your comments. We have added the HPLC and MS traces of all used peptides in the supplemental materials (Supplementary data: HPLC and MS traces of all used peptides).

We are sorry that we don't have suitable HPLC column to characterize the ligation reaction. In addition, ELISA is simpler and more convenient compared to HPLC. The throughput of ELISA is higher since dozens, even hundreds of samples can be taken simultaneously, and can be finished within 2 hours. However, HPLC is capable of analyzing only one sample at a time, with each sample requiring approximately 30 minutes for analysis, which make it time-consuming and inefficient.

3) Is the reaction also reversible? As the authors used 5-fold excess of the nucleophile, can it be further reduced? The authors mention in line 207 that “the newly producing ligating tripeptide “Asn-Arg-Leu” motif of ligating product could be less susceptible to hydrolysis”, why?

Re: Thank you very much for your comments. We think that the reaction is reversible since the product yield increases with the concentration of the nucleophile (see detailed

in Figure R3, Page 10). The nucleophile can be further reduced, but the product will be reduced also (Figure R3, Page 10).

The statement in the original manuscript suggesting that “the newly producing ligating tripeptide “Asn-Arg-Leu” motif of ligating product could be less susceptible to hydrolysis” was merely speculative and lacked sufficient evidence; we have removed the corresponding statement in the revised manuscript.

4) More rigorous kinetic studies should be performed to show the high activity of the enzyme, e.g., at different enzyme loading/concentration, and the reaction time and yield are all important information. Authors should refer to ref. 17 for a more complete characterization.

Re: Thank you very much for your comments. Reviewer #2 has also raised similar questions. According to your suggestions, we optimized some conditions, reconducted the FRET-based kinetic experiment at different concentration of enzyme and substrate, and supplemented the reaction time in the Methods section. The rising of catalytic rate was more prominent. The results showed that the K_{cat}/K_m of OaAL2 toward recognition motif “Asn-Ala-Leu” and nucleophiles “Arg-Leu” with 1 mM Fe^{3+} is superior to the previously reported “Asn-Gly-Leu” and “Gly-Leu”. And we are sorry that the turnover rate was miscalculated in the first draft, and we have corrected the calculation in the revised manuscript. Please see details in the response to comment 3 of reviewer #2 (Page 20-23).

5) In the manuscript, line 255, the authors claimed that “OaAL2 has a K_{cat}/K_m value 30 and 618 times faster than butelase-1 and AEP(Cys247Ala), respectively.”. The “30-

times” seems to be inconsistent with the results in Figure 5E and the description in line 267: The K_{cat}/K_m of OaAL2 toward recognition motif “Asn-Ala-Leu” and nucleophiles “Arg-Leu” is 16 times faster than the reported fastest butelase 1 toward “Asn-His-Val” and “Gly-Ile”. Line 234, “The K_{cat} value of OaAL2 is 158.4 s⁻¹, approximately 17 times faster than butelase-1[17], the fastest ligase reported till now (Figure 5D-E).” The “17 times” is also inconsistent with the results shown in Figure 5E.

Re: We are sorry for our careless mistakes and sincerely thank the reviewer for the thoughtful and detailed review to help us to make this article more accurate. Indeed, the “30 times” is a numerical error. Since the FRET-based kinetic studies reconducted in the revised manuscript (refer to response to comment 3 of reviewer #2, page 20-23), the kinetic parameters changed (Figure 5C-E in the revised manuscript). We have updated the results in Figure 5.

Line 325-330: “Notably, the K_{cat}/K_m value of OaAL2 towards the substrate recognition motif “Asn-Ala-Leu” and nucleophiles “Arg-Leu” is was approximately 24 times higher than OaAL2 towards the previously reported “Asn-Gly-Leu” and “Gly-Leu”, and 70 times higher than the ultrafast variant AEP(Cys247Ala), as well as 2 times higher than butelase-1 [15], the most efficient ligase reported to date (Figure 5D-E).”

Line 395-399: “The K_{cat}/K_m of OaAL2 toward superior substrate recognition motif “Asn-Ala-Leu” and nucleophiles “Arg-Leu” is 24 times higher than the previously reported “Asn-Gly-Leu” and “Gly-Leu”, and 70 times higher than the ultrafast variant AEP(Cys247Ala), as well as 2 times higher than the reported fastest butelase-1 toward “Asn-His-Val” and “Gly-Ile”. ”

6) It is interesting that Fe³⁺ facilitates the ligating activity of OaAL2 significantly. Were all ligations in this work carried out in the buffer containing Fe³⁺? If so, the concentration should be stated; if not, why? And maybe the effect of Fe³⁺ in different concentrations could be explored to further optimize the ligation activity?

Re: Thank you very much for your comments. The assays presented in Figure 3D and Figure 5 were conducted in a buffer containing Fe³⁺. The concentration of Fe³⁺ is stated in the figure legends. The pH and metal ions preference profile of OaAL2 were conducted simultaneously, so Fe³⁺ was not utilized in the pH scanning assays. In the forth part of the Results (Substrate and nucleophile specificity of OaAL2), Fe³⁺ was also not included, because we want to solely highlight the contribution of substrate and nucleophile profiles. According to your comment, the effect of Fe³⁺ was revalidated based on the best substrate (Asn-Ala-Leu, P1-P1'-P2') and nucleophile peptide (Arg-Leu, P1''-P2''), and its concentration was optimized. The result indicated that a moderate concentration of Fe³⁺ facilitated the yield of ligating product, with 1 mM performing best (Figure R9, page 21). These findings have now been included in Figure 4F of the revised manuscript.

Line 282-286: “Based on the optimal substrate of Asn-Ala-Leu (P1-P1'-P2') and nucleophile peptide of Arg-Leu (P1''-P2''), the effect of Fe³⁺ were further validated and the optimum concentration of Fe³⁺ were investigated. The result demonstrated that a moderate concentration of Fe³⁺ enhances the yield of ligating product, with 1 mM proving to be the most effective (Figure 4F).”

Below are some other minor issues:

1) The English of the manuscript is very poor and should be checked thoroughly? E.g., Line 29 “rational designed” should be “rationally designed”; Line 80, “sharpening” is not the correct word here....

Re: We apologize for the poor language of our manuscript. We have now worked on both language and readability and have also involved native English speaker for language corrections. We really hope that the flow and language level have been substantially improved.

2) The phrase “butelase 1” is not consistent, sometimes as “butelase1,” or “butelase-1”

Re: We sincerely thank the reviewer for careful reading. As suggested, we have corrected them to “butelase-1” consistently in the revised manuscript.

3) Line 126 “Nucleophiles” should be “nucleophiles”.

Re: We are so really sorry for this mistake. Thank you for your reminder. We have corrected it in the revised manuscript.

Line 164-165: “Another peptide having N-terminal residues “Gly-Leu” (GL- (biotin labeled) peptide) was utilized as the nucleophile.”

Reviewers' comments:

Reviewer #1 (Remarks to the Author):

Dear Prof Ge and co-workers,

It is satisfying to see how you put in extra hard work in verifying your results and addressing our comments to your original manuscript. Definitely there are vast improvements everywhere and the flow of the story is much clearer now, compared to its previous form. I appreciate your persistence and enthusiasm.

To my opinion, the most critical issues of the study still remain unsolved.

1. Given a new name to a truncated construct. In general, in biochemistry, fellow researchers would refer to the new boundary and keep its original name. The term 'ligase' has been given to this enzyme multiple times in the past years, and we all know its ability to ligate soluble proteins. Please show respect to your predecessors in this field.

2. 'rational' design. When you refer to rationally design something, usually there is a hypothesis and there is a systematic evaluation of that guideline. The full substrate specificity of OaAEP1-C247A was published in a public patent claim in 2017. (<https://patents.google.com/patent/US11795488B2/en>) The results in this manuscript does not enclosed a logic rationale to present why the described condition shall be more active, nor present any new specificity data.

3. Ligation 'efficiency'. Based on Figure R5, the intensity of claimed 1uM ligase and 4.5 uM ligated product, does not match the intensity of the bands, even considering the size difference. (as a comparison, please look at figure 4B of the 2017 JACS oaAEP1 paper) The protein ligation shall be crystal clear, and from nowhere we can recognise superior ligase activity from this truncated form that claimed by the authors. Is it not possible to obtain better results or it is simply not active as claimed?

We would love to see new progress in this field, however, we prefer to see solid and reproducible results that would help to move our understanding forward. I would only endorse this manuscript when sincere and accurate descriptions were disclosed. In fact, it is completely not necessary to report 'superior' data, since we are engaging in discussing potential better ways to prepare enzymes.

Reviewer #3 (Remarks to the Author):

The authors have revised the manuscript properly and all my previous concerns have been addressed. In the opinion of this reviewer, the manuscript can be accepted for publication in this journal.

Line to line responses to the reviewer's comments

Reviewer #1 (Remarks to the Author):

Dear Prof Ge and co-workers,

It is satisfying to see how you put in extra hard work in verifying your results and addressing our comments to your original manuscript. Definitely, there are vast improvements everywhere and the flow of the story is much clearer now, compared to its previous form. I appreciate your persistence and enthusiasm.

Re: Thanks very much for taking your time to review this manuscript and giving your positive comments. We are grateful to the reviewers for your constructive feedback that helped us enhance the quality of our manuscript.

To my opinion, the most critical issues of the study still remain unsolved.

1. Given a new name to a truncated construct. In general, in biochemistry, fellow researchers would refer to the new boundary and keep its original name. The term 'ligase' has been given to this enzyme multiple times in the past years, and we all know its ability to ligate soluble proteins. Please show respect to your predecessors in this field.

Re: Thank you for your reminder that referring to the new boundary and keeping its original name of the proposed ligase in this study to show respect to the predecessors in this field. According to the four OaAEP1-C247A truncations, we have modified the name to “OaAEP1-C247A-aa24-351/aa55-351/aa24-325/aa55-325”, and all the related descriptions in the revised manuscript have been corrected. Thanks very much again.

2. 'rational' design. When you refer to rationally design something, usually there is a hypothesis and there is a systematic evaluation of that guideline. The full substrate specificity of OaAEP1-C247A was published in a public patent claim in 2017. (<https://patents.google.com/patent/US11795488B2/en>) The results in this manuscript does not enclosed a logic rationale to present why the described condition shall be more active, nor present any new specificity data.

Re: Thank you for your professional comments. We greatly appreciate the opportunity to explain that when we refer to "rational design" in our article, it is indeed with specific regard to the enzyme itself. Based on the crystal structure of OaAEP1-C247A, we ventured a hypothesis that the N-terminal domain aa24-54 and C-terminal domain aa326-351 may play crucial in the ligase structure and function. This was subsequently validated by conducting a truncated expression of the enzyme to ascertain its activity, which demonstrated the critical importance of the C-terminal domain aa326-351 for ligase activity, whereas N-terminal domain aa24-54 would decrease this activity. Furthermore, with respect to substrate recognition, our investigative efforts were directed towards validating the consistency of substrate specificity between the directly expressed truncated ligase in this study and the acid-activation ligase, meanwhile discovering the most effective combination of substrate and nucleophilic motif. Our focus was thus not primarily on the specificity of substrate recognition of OaAEP1-C247A. We sincerely hope that our elaboration here meets with clarity.

3. Ligation 'efficiency'. Based on Figure R5, the intensity of claimed 1uM ligase and 4.5 uM ligated product, does not match the intensity of the bands, even considering the size difference. (as a comparison, please look at figure 4B of the 2017 JACS oaAEP1 paper) The protein ligation shall be crystal clear, and from nowhere we can recognise superior ligase activity from this truncated form that claimed by the authors. Is it not possible to obtain better results or it is simply not active as claimed?

We would love to see new progress in this field, however, we prefer to see solid and reproducible results that would help to move our understanding forward. I would only endorse this manuscript when sincere and accurate descriptions were disclosed. In fact, it is completely not necessary to report 'superior' data, since we are engaging in discussing potential better ways to prepare enzymes.

Re: We are sincerely grateful for your expert comments and for the meticulous attention you have devoted to our manuscript. We appreciate the opportunity to clarify, with all due respect, that we overlooked the difference in band intensity between ligation

product and ligase, but our study endeavors to contribute beyond merely reporting superior data.

According to your description, our understanding is that you believe that in the experiment, when 5 μM of C50-NAL was used, with a ligation efficiency of 90% for the ligase, it should produce 4.5 μM of ligation products theoretically. It needs to be clarified that the description in the manuscript, “90% of C50-NAL was ligated with RL- (biotin labeled) polyXXXK peptide”, does not imply that a 90% yield of soluble ligation products. We identified sediments in the electrophoresis wells after ligation (indicated by a red triangle in Figure R1), a phenomenon not present in the controls (Figure R1). It demonstrated that this sedimentation, presumably from the ligation products, could potentially contribute to the observed reduction in band intensity of the ligation product, which resulted from the characteristics of the ligated peptides, rather than the ligase itself (Figure R1). Additionally, we overlooked the difference in band intensity between ligated product and ligase. We noticed that the discrepancy in band intensity was apparent between ligase and C50-NAL even before the ligation process, leading us to speculate that it might be attributable to a suboptimal staining effect of small peptides. To validate the reproducibility of this experiment, we revisited the assay (Figure R2), conducting another set of electrophoresis for the ligation product (Lane 4) alongside our controls (Lanes 2-4). These controls comprise ligase (1 μM , Lane 1), C50-NAL (5 μM , Lane 2), RL- (biotin-labeled) polyXXXK (50 μM , Lane 3), respectively. We once again noticed the variance in band intensity between the ligase and C50-NAL. The ligation efficiency was indeed repeatable.

Our aim is to convey this aspect with greater precision, and we appreciate the opportunity to clarify these points in response to your valuable feedback.

Figure R1. Ligation products precipitated after ligation. Red triangles indicate the precipitated ligation products.

Figure R2. SDS-PAGE analysis of ligation substrates and products of OaAEP1-C247A-aa55-351. Lane 1: 1 μ M OaAEP1-C247A-aa55-351; Lane 2: 5 μ M C50-NAL; Lane 3: 50 μ M RL- (biotin-labeled) polyXK; Lane 4: ligation products.

Reviewer #3 (Remarks to the Author):

The authors have revised the manuscript properly and all my previous concerns have been addressed. In the opinion of this reviewer, the manuscript can be accepted for publication in this journal.

Re: I would like to express my heartfelt gratitude for your positive feedback and for considering our manuscript for publication. Your constructive comments have been invaluable to the refinement of our work, and we are deeply appreciative of the time and effort you have invested in reviewing our submission. Thank you once again for your support and encouragement.

REVIEWERS' COMMENTS:

Reviewer #1 (Remarks to the Author):

We thank the authors putting in additional efforts updating the manuscripts according to the comments. Now the revision has clarified the core message, that a refined OaAEP1b-C247A boundary tagged with a TrxA tag, can be directly expressed, purified and shown protein ligase activity to a measurable extent. This would potentially inspire further endeavors in the realm of enzyme engineering within related field.